# Predicting effective microRNA target sites in mammalian mRNAs

**Vikram Agarwal[1,2,3], George W Bell[4], Jin-Wu Nam[1,2,5], David P Bartel[1,2]***

[1]Howard Hughes Medical Institute, Whitehead Institute for Biomedical Research, Cambridge, United States; [2]Department of Biology, Massachusetts Institute of Technology, Cambridge, United States; [3]Computational and Systems Biology Program, Massachusetts Institute of Technology, Cambridge, United States; [4]Bioinformatics and Research Computing, Whitehead Institute for Biomedical Research, Cambridge, United States; [5]Department of Life Science, College of Natural Sciences, Hanyang University, Seoul, Republic of Korea

**Abstract** MicroRNA targets are often recognized through pairing between the miRNA seed region and complementary sites within target mRNAs, but not all of these canonical sites are equally effective, and both computational and in vivo UV-crosslinking approaches suggest that many mRNAs are targeted through non-canonical interactions. Here, we show that recently reported non-canonical sites do not mediate repression despite binding the miRNA, which indicates that the vast majority of functional sites are canonical. Accordingly, we developed an improved quantitative model of canonical targeting, using a compendium of experimental datasets that we pre-processed to minimize confounding biases. This model, which considers site type and another 14 features to predict the most effectively targeted mRNAs, performed significantly better than existing models and was as informative as the best high-throughput in vivo crosslinking approaches. It drives the latest version of TargetScan (v7.0; targetscan.org), thereby providing a valuable resource for placing miRNAs into gene-regulatory networks.

*For correspondence: dbartel@wi.mit.edu

**Competing interests:** The authors declare that no competing interests exist.

**Reviewing editor**: Elisa Izaurralde, Max Planck Institute Development Biology, Germany

## Introduction

MicroRNAs (miRNAs) are ~22-nt RNAs that mediate post-transcriptional gene repression (*Bartel, 2004*). Bound with an Argonaute protein to form a silencing complex, miRNAs function as sequence-specific guides, directing the silencing complex to transcripts, primarily through Watson–Crick pairing between the miRNA seed (miRNA nucleotides 2–7) and complementary sites within the 3′ untranslated regions (3′ UTRs) of target RNAs (*Lewis et al., 2005*; *Bartel, 2009*). The miRNAs conserved to fish have been grouped into 87 families, each with a unique seed region. On average, each of these families has >400 conserved targeting interactions, and together these interactions involve most mammalian mRNAs (*Friedman et al., 2009*). In addition, many nonconserved interactions also function to reduce mRNA levels and protein output (*Farh et al., 2005*; *Krutzfeldt et al., 2005*; *Lim et al., 2005*; *Baek et al., 2008*; *Selbach et al., 2008*). Accordingly, miRNAs have been implicated in a wide range of biological processes in worms, flies, and mammals (*Kloosterman and Plasterk, 2006*; *Bushati and Cohen, 2007*; *Stefani and Slack, 2008*). Critical for understanding miRNA biology is the accurate prediction of miRNA–target interactions. Although numerous advances have been made, accurate and specific target predictions remain a challenge.

Analysis of preferentially conserved miRNA-pairing motifs within 3′ UTRs has led to the identification of several classes of target sites (*Bartel, 2009*). The most effective canonical site types, listed in order of decreasing preferential conservation and efficacy, are the 8mer site (Watson–Crick match to miRNA positions 2–8 with an A opposite position 1 [*Lewis et al., 2005*]), 7mer-m8 site

**eLife digest** Proteins are built by using the information contained in molecules of messenger RNA (mRNA). Cells have several ways of controlling the amounts of different proteins they make. For example, a so-called 'microRNA' molecule can bind to an mRNA molecule to cause it to be more rapidly degraded and less efficiently used, thereby reducing the amount of protein built from that mRNA. Indeed, microRNAs are thought to help control the amount of protein made from most human genes, and biologists are working to predict the amount of control imparted by each microRNA on each of its mRNA targets.

All RNA molecules are made up of a sequence of bases, each commonly known by a single letter—'A', 'U', 'C' or 'G'. These bases can each pair up with one specific other base—'A' pairs with 'U', and 'C' pairs with 'G'. To direct the repression of an mRNA molecule, a region of the microRNA known as a 'seed' binds to a complementary sequence in the target mRNA. 'Canonical sites' are regions in the mRNA that contain the exact sequence of partner bases for the bases in the microRNA seed. Some canonical sites are more effective at mRNA control than others. 'Non-canonical sites' also exist in which the pairing between the microRNA seed and mRNA does not completely match. Previous work has suggested that many non-canonical sites can also control mRNA degradation and usage.

Agarwal et al. first used large experimental datasets from many sources to investigate microRNA activity in more detail. As expected, when mRNAs had canonical sites that matched the microRNA, mRNA levels and usage tended to drop. However, no effect was observed when the mRNAs only had recently identified non-canonical sites. This suggests that microRNAs primarily bind to canonical sites to control protein production.

Based on these results, Agarwal et al. further developed a statistical model that predicts the effects of microRNAs binding to canonical sites. The updated model considers 14 different features of the microRNA, microRNA site, or mRNA—including the mRNA sequence around the site—to predict which sites within mRNAs are most effectively targeted by microRNAs. Tests showed that Agarwal et al.'s model was as good as experimental approaches at identifying the effective target sites, and was better than existing computational models.

The model has been used to power the latest version of a freely available resource called TargetScan, and so could prove a valuable resource for researchers investigating the many important roles of microRNAs in controlling protein production.

(position 2–8 match [*Brennecke et al., 2005*; *Krek et al., 2005*; *Lewis et al., 2005*]), and 7mer-A1 site (position 2–7 match with an A opposite position 1 [*Lewis et al., 2005*]). Experiments have confirmed that the preference for an adenosine opposite position 1 is independent of the miRNA nucleotide identity (*Grimson et al., 2007*; *Nielsen et al., 2007*; *Baek et al., 2008*) and due to the specific recognition of the target adenosine within a binding pocket of Argonaute (*Schirle et al., 2014*). Two other canonical site types, each associated with weaker preferential conservation and much lower efficacy (*Friedman et al., 2009*), are the 6mer (position 2–7 match [*Lewis et al., 2005*]) and offset-6mer (position 3–8 match [*Friedman et al., 2009*]). Pairing to the 3′ end of the miRNA can supplement canonical sites, although evidence for the use of this 3′-supplementary pairing is observed for no more than 5% of the seed-matched sites (*Brennecke et al., 2005*; *Lewis et al., 2005*; *Grimson et al., 2007*; *Friedman et al., 2009*).

Some effective sites lack canonical seed pairing. For example, very extensive pairing to the 3′ region of the miRNA can compensate for a wobble or mismatch to one of the seed positions (*Doench and Sharp, 2004*; *Brennecke et al., 2005*; *Bartel, 2009*), as exemplified by the two *let-7* sites within the 3′ UTR of *Caenorhabditis elegans lin-41* (*Reinhart et al., 2000*). Although these 3′-supplementary sites can be detected above background when searching for preferentially conserved pairing configurations, they are exceedingly rare, with conserved miRNA families in mammals and nematodes each averaging <1 preferentially conserved 3′-supplementary site (*Friedman et al., 2009*). Other relatively rare, yet effective sites include centered sites, which have 11–12 contiguous Watson–Crick pairs to the center of the miRNA (*Shin et al., 2010*), and cleavage sites, which have the very extensive pairing required for Argonaute-catalyzed slicing of the mRNA (*Yekta et al., 2004*; *Davis et al., 2005*; *Karginov et al., 2010*; *Shin et al., 2010*). The existence of additional, still-to-be-characterized types

of non-canonical sites is suggested by the large number of mRNA regions that crosslink to the silencing complex in vivo yet lack known site types matching the cognate miRNA (*Chi et al., 2012*; *Loeb et al., 2012*; *Helwak et al., 2013*; *Khorshid et al., 2013*; *Grosswendt et al., 2014*).

With the prediction of hundreds of conserved targets for most mammalian miRNAs (and even more nonconserved targets), knowing which targets are expected to be most responsive to each miRNA provides important information for both large-scale network analyses and detailed experimental follow-up. As previously mentioned, the type of site (e.g., whether the site is an 8mer or a 7mer-A1) strongly influences the efficacy of repression. The number of sites also influences efficacy, with each additional site typically acting independently to impart additional repression (*Grimson et al., 2007*; *Nielsen et al., 2007*), although sites between 8–40 nt of each other tend to act cooperatively, and those < 8 nt of each other act competitively (*Grimson et al., 2007*). Additional features of site context help explain why a given site (e.g., a 7mer-m8 site to miR-1) can be more effective in one 3′ UTR than it is in another. These features include the positioning of the site outside of the path of the ribosome (which includes the first 15 nt of the 3′ UTR [*Grimson et al., 2007*]) and the positioning of the site within 3′-UTR segments that are more accessible to the silencing complex, as measured by either high local AU content (*Grimson et al., 2007*; *Nielsen et al., 2007*), high AU content of the entire 3′ UTR (*Robins and Press, 2005*; *Hausser et al., 2009*), shorter distance from a 3′-UTR terminus (*Gaidatzis et al., 2007*; *Grimson et al., 2007*; *Majoros and Ohler, 2007*), shorter 3′-UTR length (*Hausser et al., 2009*; *Betel et al., 2010*; *Wen et al., 2011*; *Reczko et al., 2012*), or less stable predicted competing secondary structure (*Robins et al., 2005*; *Ameres et al., 2007*; *Kertesz et al., 2007*; *Long et al., 2007*; *Tafer et al., 2008*). Conserved sites are also more effective, in part because they tend to reside in more favorable contexts (*Grimson et al., 2007*; *Nielsen et al., 2007*). Features of the miRNA can also influence site efficacy, with sites being more effective if the miRNA has lower target-site abundance (TA) within the transcriptome (*Arvey et al., 2010*; *Garcia et al., 2011*) and stronger predicted seed-pairing stability (SPS) (*Garcia et al., 2011*).

Multiple features can be considered together to build quantitative models of targeting efficacy (*Grimson et al., 2007*; *Nielsen et al., 2007*; *Wang and El Naqa, 2008*; *Betel et al., 2010*; *Liu et al., 2010*; *Garcia et al., 2011*; *Wen et al., 2011*; *Reczko et al., 2012*; *Vejnar and Zdobnov, 2012*; *Marin et al., 2013*; *Gumienny and Zavolan, 2015*). Our recent model, called the context-plus (context+) model, considers the features of our original context scores (i.e., site type, 3′-supplementary pairing, local AU content, and distance from the closest 3′-UTR end [*Grimson et al., 2007*]), plus two miRNA features (TA and SPS [*Garcia et al., 2011*]). Although the context+ model was trained using multiple regression on 74 high-throughput datasets, the features used to distinguish effective sites (the three features of the original context scores) were identified using only 11 datasets, implying that additional features might be identified through analysis of the additional datasets.

Here, we examined the function of non-canonical binding sites identified in recent studies and found that mRNAs with these sites are not more repressed than mRNAs without sites, despite compelling evidence that many of these noncanocial sites bind the silencing complex in vivo. This finding justified a focus on the statistical modeling of canonical, seed-matched sites within 3′ UTRs, which mediate the vast majority of repression that can be predicted with current methods. To this end, we pre-processed the 74 datasets to minimize confounding biases and then used stepwise regression to identify the most informative features from a large set of potential targeting features. This approach unbiasedly selected 14 features, which were combined to develop the context++ model of miRNA targeting efficacy. The context++ model was more predictive than any published model and at least as predictive as the most informative in vivo crosslinking approaches. As the engine powering the latest version of TargetScan (v7.0; targetscan.org), this model provides a valuable resource for placing the miRNAs of human, mouse, zebrafish, and other vertebrate species into their respective gene-regulatory networks.

## Results

### Inefficacy of recently reported non-canonical binding sites

Several high-throughput crosslinking-immunoprecipitation (CLIP) approaches have been applied to identify sites that bind Argonaute in vivo (*Chi et al., 2009*; *Hafner et al., 2010*; *Helwak et al., 2013*; *Grosswendt et al., 2014*). These experiments all observe significant enrichment for cognate seed-matched sites in the vicinity of the crosslinks, which validates their ability to detect authentic sites.

Despite this enrichment, some crosslinks do not correspond to canonical sites to the relevant miRNAs, raising the prospect that these results might reveal novel types of non-canonical binding that could mediate repression. Indeed, five studies have reported crosslinking to non-canonical binding sites proposed to mediate repression (*Chi et al., 2012*; *Loeb et al., 2012*; *Helwak et al., 2013*; *Khorshid et al., 2013*; *Grosswendt et al., 2014*). In addition, another biochemical study has reported the identification of non-canonical sites without using any crosslinking (*Tan et al., 2014*). Reasoning that these experimental datasets might provide a resource for defining of novel types of sites to be used in target prediction, we re-examined the functionality of these sites in mediating target mRNA repression.

We first examined the efficacy of 'nucleation-bulge' sites (*Chi et al., 2012*), which were identified from analysis of differential CLIP (dCLIP) results reporting the clusters that appear in the presence of miR-124 (*Chi et al., 2009*). Nucleation-bulge sites consist of 8 nt motifs paired to positions 2–8 of their cognate miRNA seed, with the nucleotide opposing position 6 protruding as a bulge but sharing Watson-Crick complementarity to miRNA position 6. Meta-analyses of miRNA and small-RNA transfection datasets revealed significant repression of mRNAs with the canonical site types but found no evidence for repression of mRNAs that contain nucleation-bulge sites but lack perfectly paired seed-matched sites in their 3′ UTRs (*Figure 1—figure supplement 1A,B*). Reasoning that the nucleation-bulge site might be only marginally effective, we examined the early zebrafish embryo with and without Dicer, analyzing the targeting by miR-430, the most highly expressed miRNA of the early embryo. Even in this system, one of the most sensitive systems for detecting the effects of targeting (where a robust repression is observed for mRNAs with only a single 6mer or offset-6mer sites to miR-430), we observed no evidence for repression of mRNAs with nucleation-bulge sites to miR-430 (*Figure 1A*, *Figure 1—figure supplement 1C*, and *Figure 1—figure supplement 4A*). Because the nucleation-bulge sites were originally identified and characterized as sites to miR-124, we next tried focusing on only miR-124–mediated repression. However, even in this more limited context, the mRNAs with nucleation-bulge sites were no more repressed than mRNAs without sites (*Figure 1—figure supplement 1D–F*).

Another study examined the response of 32 mRNAs that lack canonical miR-155 sites yet crosslink to Argonaute in wild-type T cells but not T cells isolated from miR-155 knockout mice (*Loeb et al., 2012*). As previously observed, we found that the levels of these mRNAs tended to increase in T cells lacking miR-155 (*Figure 1B*). However, a closer look at the distribution of mRNA fold changes between wild-type and knockout cells revealed a pattern not normally observed for mRNAs with a functional site type. As illustrated for the mRNAs with canonical sites (including those supported by CLIP), when a miRNA is knocked out, the cumulative distribution of fold changes for mRNAs with functional site types diverges most from the no-site distribution at the top of the curve, which represents the most strongly derepressed mRNAs (*Figure 1B*). However, for the mRNAs harboring non-canonical miR-155 sites, the distribution of fold changes converged with the no-site distribution at the top of the curve (*Figure 1B*), raising doubt as to whether non-canonical binding of these mRNAs mediates repression. To investigate these mRNAs further, we examined their response to the miR-155 loss in helper T cell subtypes 1 and 2 ($T_h$1 and $T_h$2, respectively) and B cells, which are other lymphocytic cells in which significant derepression of miR-155 targets is observed in cells lacking miR-155 (*Rodriguez et al., 2007*; *Eichhorn et al., 2014*). In contrast to mRNAs with canonical sites, the mRNAs with non-canonical sites showed no evidence of derepression in the knockout cells of each of these cell types, which reinforced the conclusion that non-canonical binding of miR-155 does not lead to repression of these mRNAs (*Figure 1C* and *Figure 1—figure supplement 2*).

We next probed the functionality of non-canonical interactions identified by CLASH (crosslinking, ligation, and sequencing of hybrids), a high-throughput technique that generates miRNA–mRNA chimeras, which each identify a miRNA and the mRNA region that it binds (*Helwak et al., 2013*). As previously observed, mRNAs with CLASH-identified non-canonical interactions involving miR-92 tended to be slightly up-regulated upon knockdown of miR-92 in HEK293 cells (*Figure 1D*). However, a closer look at the mRNA fold-change distributions again revealed a pattern not typically observed for mRNAs with a functional site type, with convergence with the no-site distribution in the region expected to be most divergent. Therefore, we examined a second dataset monitoring mRNA changes after knocking down miR-92 and other miRNAs in HEK293 cells (*Hafner et al., 2010*). As reported recently (*Wang, 2014*), the slight up-regulation observed for mRNAs with CLASH-identified non-canonical interactions in the original dataset was not reproducible in the second dataset (*Figure 1E*).

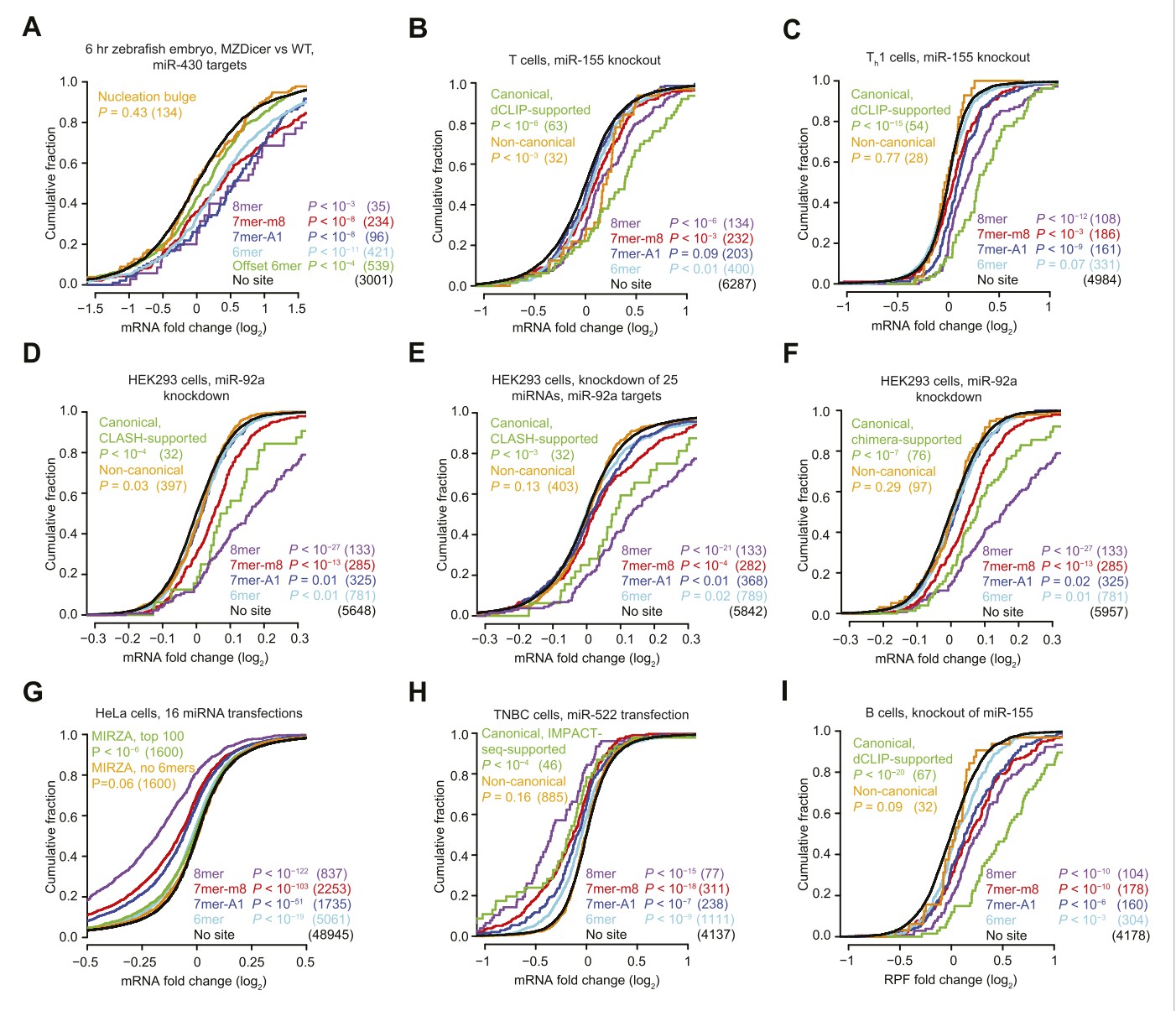

**Figure 1**. Inefficacy of recently reported non-canonical sites. (**A**) Response of mRNAs to the loss of miRNAs, comparing mRNAs that contain either a canonical or nucleation-bulge site to miR-430 to those that do not contain a miR-430 site. Plotted are cumulative distributions of mRNA fold changes observed when comparing embryos that lack miRNAs (MZDicer) to those that have miRNAs (WT), focusing on mRNAs possessing a single site of the indicated type in their 3′ UTR. Similarity of site-containing distributions to the no-site distribution was tested (one-sided Kolmogorov–Smirnov [K–S] test, $P$ values); the number of mRNAs analyzed in each category is listed in parentheses. See also *Figure 1—figure supplement 1C* and *Figure 1—figure supplement 4A*. (**B** and **C**) Response of mRNAs to the loss of miR-155, focusing on mRNAs that contain either a single canonical or ≥1 CLIP-supported non-canonical site to miR-155. These panels are as in (**A**), but compare fold changes for mRNAs with the indicated site type following genetic ablation of *mir-155* in either T cells (**B**) or T$_h$1 cells (**C**). See also *Figure 1—figure supplement 2*. (**D** and **E**) Response of mRNAs to the knockdown of miR-92a, focusing on mRNAs that contain either a single canonical or ≥1 CLASH-identified non-canonical site to miR-92a. These panels are as in (**A**), except CLASH-supported non-canonical sites were the same as those defined previously (*Helwak et al., 2013*) and thus were permitted to reside in any region of the mature mRNA, and these panels compare fold changes for mRNAs with the indicated site type following either knockdown of miR-92a (**D**) or combined knockdown of miR-92a and 24 other miRNAs (**E**) in HEK293 cells. See also *Figure 1—figure supplement 3A,B*. (**F**) As in (**D**), but focusing on mRNAs that contain ≥1 chimera-identified site. See also *Figure 1—figure supplement 3C–E* and *Figure 1—figure supplement 4B*. (**G**) Response of mRNAs to the transfection of 16 miRNAs, focusing on mRNAs that contain either a canonical or MIRZA-predicted non-canonical site. This panel is as in (**A**), but compares the fold changes for mRNAs with the indicated site type after introducing miRNAs, aggregating results from 16 individual transfection datasets. Fold changes are plotted for the top 100 non-canonical predictions for each of 16 miRNAs compiled either before (MIRZA, top 100) or after (MIRZA, no 6mers) removing mRNAs containing 6mer or offset-6mer 3′-UTR sites. (**H**) Response of mRNAs to a transfection of miR-522, focusing on mRNAs that contain

*Figure 1. continued on next page*

*Figure 1. Continued*

either a single canonical or ≥1 IMPACT-seq–supported non-canonical site to miR-522. These panels are as in (**A**), except IMPACT-seq–supported non-canonical sites were the same as those defined previously (*Tan et al., 2014*) and thus were permitted in any region of the mature mRNA. (**I**) Response of ribosomes to the loss of miR-155, focusing on mRNAs that contain either a single canonical or ≥1 CLIP-supported non-canonical site to miR-155. This panel is as in (**B** and **C**) but compares the response of mRNAs using ribosome-footprint profiling (*Eichhorn et al., 2014*) after genetic ablation of *mir-155* in B cells. Ribosome-footprint profiling captures changes in both mRNA stability and translational efficiency through the high-throughput sequencing of ribosome-protected mRNA fragments (RPFs).

The following figure supplements are available for figure 1:

**Figure supplement 1**. Inefficacy of nucleation-bulge sites.

**Figure supplement 2**. Inefficacy of CLIP-supported non-canonical miR-155 sites.

**Figure supplement 3**. Inefficacy of CLASH- and chimera-supported non-canonical sites.

**Figure supplement 4**. Inefficacy of non-canonical sites in mediating translational repression.

**Figure supplement 5**. Re-evaluating conservation of chimera-supported non-canonical sites.

Moreover, mRNAs with non-canonical interactions to other miRNAs showed no sign of derepression when the cognate miRNAs were knocked down (*Figure 1—figure supplement 3A*). To mirror the original analyses of CLASH-identified interactions (*Helwak et al., 2013*), our analyses included sites located in any region of the mature mRNA (*Figure 1D,E* and *Figure 1—figure supplement 3A*). No significant difference from the no-site control distribution was observed when restricting our analysis to mRNAs with CLASH-identified non-canonical sites in their 3′ UTRs (*Figure 1—figure supplement 3B*).

Many miRNA–mRNA chimeras can also be found in standard AGO CLIP datasets, presumably generated by an endogenous ligase acting in cell lysates during workup (*Grosswendt et al., 2014*). Global experiments examining function of these interactions group the mRNAs with non-canonical interactions together with those with canonical interactions (*Grosswendt et al., 2014*), and thus the signal for function might arise from only canonical interactions. Indeed, when we re-examined the response of these mRNAs to miRNA knockdown, those with chimera-identified canonical sites tended to be derepressed, whereas those with only chimera-identified non-canonical sites did not (*Figure 1F* and *Figure 1—figure supplement 3C–E*). Although at first glance this finding might seem at odds with the elevated evolutionary conservation of chimera-identified non-canonical sites (*Grosswendt et al., 2014*), we found that this conservation signal was not smaller for the sites of less conserved miRNAs and therefore was not indicative of functional miRNA binding (*Figure 1—figure supplement 5*). Instead, the reported conservation signal might occur for the same reason that artificial siRNAs tend to target conserved regions of 3′ UTRs (*Nielsen et al., 2007*).

Next, we evaluated the response of non-canonical sites modeled by MIRZA, an algorithm that utilizes CLIP data in conjunction with a biophysical model to predict target sites (*Khorshid et al., 2013*). As noted by others (*Majoros et al., 2013*), the definition of non-canonical MIRZA sites was more expansive than that used elsewhere and did not exclude sites with canonical 6mer or offset-6mer seed matches. Indeed, when focusing on only targets without 6mer or offset-6mer seed matches, the top 100 non-canonical MIRZA targets showed no sign of efficacy (*Figure 1G*).

Finally, we examined non-canonical clusters identified by IMPACT-seq (identification of miRNA-responsive elements by pull-down and alignment of captive transcripts—sequencing), a method that sequences mRNA fragments that co-purify with a biotinylated miRNA without crosslinking (*Tan et al., 2014*). Although the mRNAs with an IMPACT-seq–supported canonical site were down-regulated upon the transfection of the cognate miRNA, those with an IMPACT-seq–supported non-canonical site responded no differently than mRNAs lacking a site (*Figure 1H*).

Collectively, the novel non-canonical sites recently identified in high-throughput CLIP and other biochemical studies imparted no detectable repression when monitoring mRNA changes. However, monitoring of only mRNA changes leaves open the possibility that these sites might still mediate

translational repression. To address this possibility, we examined ribosome-profiling and proteomic datasets, which capture repression also occurring at the level of translation, and again we found that the CLIP-identified non-canonical sites imparted no detectable repression (*Figure 1I* and *Figure 1—figure supplement 4*).

All of our analyses of experimentally identified non-canonical sites examined the ability of the sites to act in mRNAs that had no seed-matched site to the same miRNA in their 3′ UTRs. Any non-canonical site found in a 3′ UTR that also had a seed-matched site to the same miRNA was not considered because any response could be attributed to the canonical site. At first glance, excluding these co-occurring sites might seem to allow for the possibility that the experimentally identified non-canonical sites could contribute to repression when in the same 3′ UTR as a canonical site, even though they are ineffective in 3′ UTRs without canonical sites. However, in mammals, canonical sites to the same miRNA typically act independently (*Grimson et al., 2007*; *Nielsen et al., 2007*), and we have no reason to think that non-canonical sites would behave differently. More importantly, although the non-canonical sites examined were in mRNAs that had no seed-matched 3′-UTR site to the same miRNA, most were in mRNAs that had seed-matched 3′-UTR sites to other miRNAs that were highly expressed in the cells. Therefore, even if the non-canonical sites could only function when coupled to a canonical site, we still would have observed a signal for their function in our analyses.

## Confirmation that miRNAs bind to non-canonical sites despite their inefficacy

The inefficacy of recently reported non-canonical sites was surprising when considering evidence that the dCLIP clusters without cognate seed matches are nonetheless enriched for imperfect pairing to the miRNA, which would not be expected if those clusters were merely non-specific background (*Chi et al., 2012*; *Loeb et al., 2012*). Indeed, our analysis of motifs within the dCLIP clusters for miR-124 and miR-155 confirmed that those without a canonical site to the miRNA were enriched for miRNA pairing (*Figure 2A*). Although one of the motifs identified within CLIP clusters that appeared after transfection of miR-124 into HeLa cells yet lacked a canonical miR-124 site did not match the miRNA (*Figure 2—figure supplement 1C*), the top motif, as identified by MEME (*Bailey and Elkan, 1994*), had striking complementarity to the miR-124 seed region (*Figure 2A*). This human miR-124 non-canonical motif matched the 'nucleation-bulge' motif originally found for miR-124 in the mouse brain (*Chi et al., 2012*). Although the top motif identified within the subset of miR-155 dCLIP clusters that lacked a canonical site to miR-155 was not identified with confidence, it had only a single mismatch to the miR-155 seed, which would not have been expected for a motif identified by chance.

Previous analysis of CLASH-identified interactions shows that the top MEME-identified motifs usually pair to the miRNA, although for many miRNAs this pairing falls outside of the seed region (*Helwak et al., 2013*). Repeating this analysis, but focusing on only interactions without canonical sites, confirmed this result (*Figure 2B*). Applying this type of analysis to non-canonical interactions identified from miRNA–mRNA chimeras in standard AGO CLIP datasets confirmed that these interactions are also enriched for pairing to the miRNA (*Grosswendt et al., 2014*). As previously shown (*Grosswendt et al., 2014*), these interactions were more specific to the seed region than were the CLASH-identified interactions (*Figure 2B*). Comparison of all the chimera data with all the CLASH data showed that a higher fraction of the chimeras captured canonical interactions and that a higher fraction captured interactions within 3′ UTRs (*Figure 2—figure supplement 1A*). These results, implying that the chimera approach is more effective than CLASH at capturing functional sites that mediate repression, motivated a closer look at the chimera-identified interactions that lacked a canonical site, despite our finding that these interactions do not mediate repression. In the human and nematode datasets (and less so in the mouse dataset), these interactions were enriched for motifs that corresponded to non-canonical sites that paired to the miRNA seed region (*Figure 2B–C*, *Figure 2—figure supplement 1B*, and *Figure 2—figure supplement 2*). Inspection of these motifs revealed that the most enriched nucleotides typically preserved Watson–Crick pairing in a core 4–5 nts within the seed region, with tolerance to mismatches or G:U wobbles observed at varied positions, depending on the miRNA, potentially reflecting seed-specific structural or energetic features, or perhaps context-dependent biases in crosslinking or ligation.

Motifs for only a few miRNAs had a bulged nucleotide, and if a bulge was observed it was in the mRNA strand and not in the miRNA strand, as expected if the Argonaute protein imposed geometric

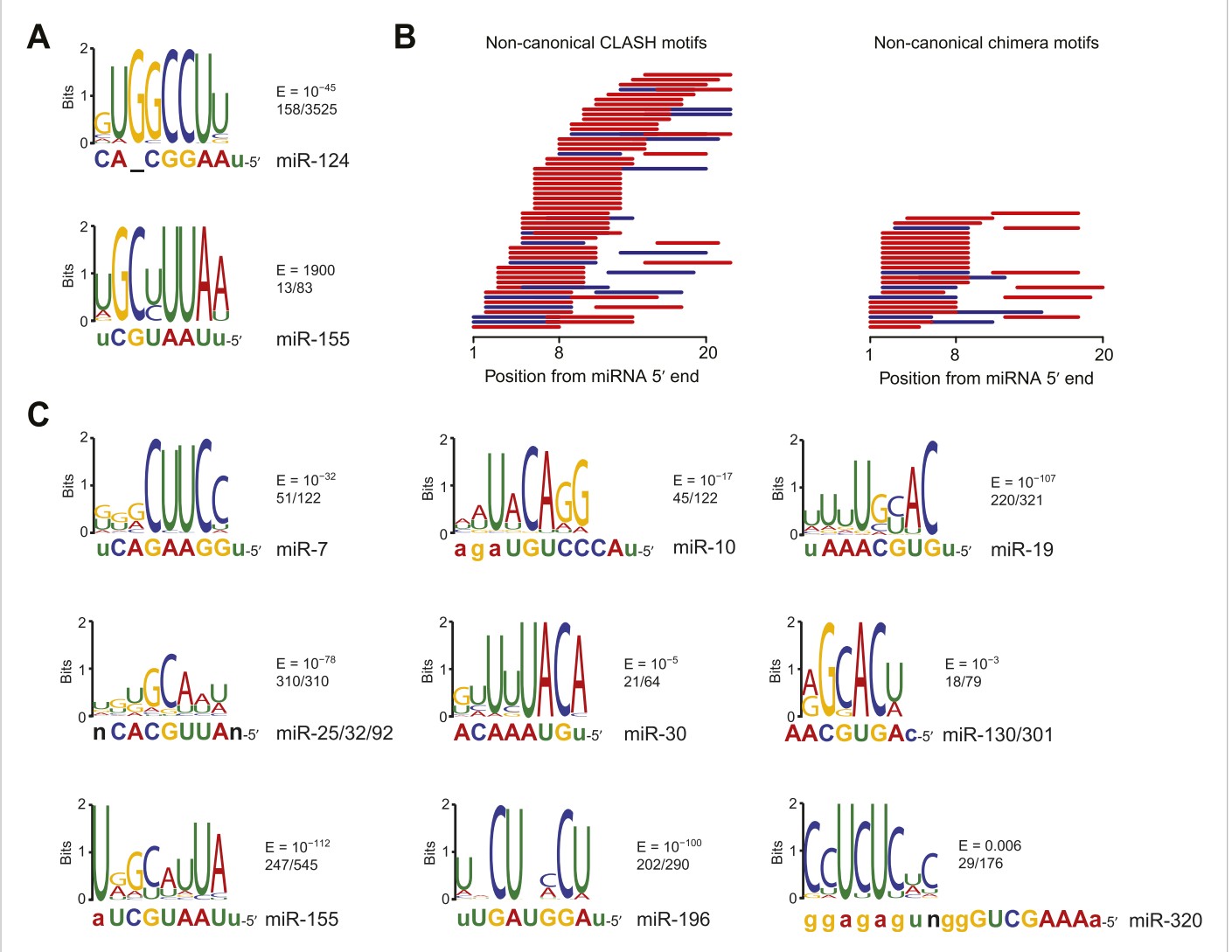

**Figure 2**. Confirmation of experimentally identified non-canonical miRNA binding sites. (**A**) Sequence logos corresponding to motifs enriched in dCLIP clusters that either appear following transfection of miR-124 into HeLa cells (*Chi et al., 2009*) (top) or disappear following knockout of miR-155 in T cells (*Loeb et al., 2012*) (bottom). Shown to the right of each logo is its E-value among clusters lacking a seed-matched or offset-6mer canonical site and the fraction of these clusters that matched the logo. Shown below each logo are the complementary regions of the cognate miRNA family, highlighting nucleotides 2–8 in capital letters. (**B**) Position of the top-ranked motif corresponding to non-canonical sites enriched in CLASH (*Helwak et al., 2013*) (left) or chimera (*Grosswendt et al., 2014*) (right) data for each human miRNA family supported by at least 50 interactions without a seed-matched or offset-6mer canonical site. For each family the most enriched logo was aligned to the reverse complement of the miRNA. In cases in which a logo mapped to multiple positions along the miRNA, the positions with the best and second best scores are indicated (red and blue, respectively). (**C**) Sequence logos of motifs enriched in chimera interactions that lack canonical sites. As in (**A**), but displaying sequence logos identified in the chimera data of panel (**B**) for a sample of nine human miRNAs. Logos identified for the other human miRNAs are also provided (*Figure 2—figure supplement 1B*). A nucleotide that differs between miRNA family members is indicated as a black 'n'.

The following figure supplements are available for figure 2:

**Figure supplement 1**. Comparison of CLASH and chimera data and identification of motifs enriched in human chimera interactions that lack canonical sites.

**Figure supplement 2**. Identification of motifs enriched in mouse and nematode chimera interactions that lack canonical sites.

constraints in the seed of the miRNA. The miR-124 nucleation-bulge site was enriched in mouse chimera interactions (*Figure 2—figure supplement 2A*), as it had been in the human and mouse dCLIP clusters (*Figure 2A*) (*Chi et al., 2012*). However, despite identification of this miR-124 interaction in datasets from two methods and two species, this style of bulged pairing was not detected for any other miRNA. Interestingly, for all other cases in which a bulge in the recognition motif was observed (human miR-33 and miR-374, and *C. elegans* miR-50 and miR-58), the bulge was between the nucleotides that paired to miRNA nucleotides 4 and 5 (*Figure 2—figure supplement 1B* and *Figure 2—figure supplement 2B*). A bulge is observed between the analogous nucleotides of validated targets of *Arabidopsis* miR398 (*Jones-Rhoades and Bartel, 2004*), whereas single-nucleotide bulges between other seed-pairing positions have not been reported in other validated plant targets. A bulge between these nucleotides is also observed in the first *let-7* site in the *C. elegans lin-41* 3′ UTR, one of the archetypal 3′-compensatory sites (*Reinhart et al., 2000*; *Bartel, 2009*). Taken together, these observations suggest that the most tolerated bulge in miRNA seed pairing is between the target nucleotides that pair to miRNA nucleotides 4 and 5.

Some motifs, particularly the more degenerate ones, were found in most of the interactions, whereas other motifs were found in only a small minority (*Figure 2C* and *Figure 2—figure supplement 1B*). We suspect that many of the interactions lacking the top-scoring motifs also involve non-canonical binding sites, some of which might function through degenerate versions of the motif that happened to have scored highest in the MEME analysis. Nonetheless, some interactions or CLIP clusters lacking the top-scoring motifs might represent background (*Friedersdorf and Keene, 2014*), and indeed a few with the motif or even with a canonical site might represent background.

In sum, our analyses of the CLIP datasets confirmed that many of the CLIP clusters and CLASH/chimera interactions lacking a seed match nonetheless capture authentic miRNA-binding sites—otherwise the top enriched motifs would not pair so often to the cognate miRNA. Despite this ability to bind the miRNA in vivo and to function in the sense that they contribute to cellular TA (*Denzler et al., 2014*), we classify the CLIP-identified non-canonical sites as non-functional with respect to repression because they showed no sign of mediating repression and no signal for miRNA-dependent conservation (*Figure 1* and *Figure 1—figure supplements 1–5*). Thus, the only known non-canonical site types that mediate repression are the 3′-supplementary, centered, and cleavage site types, which together comprise <1% of the effective sites that currently can be predicted (*Friedman et al., 2009*; *Shin et al., 2010*). Although we cannot exclude the possibility that additional types of functional non-canonical sites might exist but have not yet been characterized to the point that they can be used for miRNA target prediction (*Lal et al., 2009*), our analysis of the CLIP results justified a focus on the abundant site types that are predictive of targeting and are at least marginally functional, that is, the canonical seed-matched sites, including 6mer and offset-6mer sites.

## Improving dataset quality for model development

To identify features involved in mammalian miRNA targeting, we analyzed the results of microarray datasets reporting the mRNA changes after transfecting either a miRNA or siRNA (together referred to as small RNAs, abbreviated as sRNAs) into HeLa cells. From the published datasets, we used the set of 74 experiments that had previously been selected because each (1) had a clear signal for sRNA-based repression, (2) was acquired using the same Agilent array platform, and (3) reported on the effects of a unique seed sequence (*Garcia et al., 2011*).

Despite the differences among the 74 transfected sRNAs, mRNA fold changes of some arrays were highly correlated with those of others, which indicated that sRNA-independent effects dominated (*Figure 3A*). When all 74 datasets were compared against each other, those from either the same group of experiments (*Anderson et al., 2008*) or the same transfection protocol (*Jackson et al., 2006a*, *2006b*; *Grimson et al., 2007*) tended to cluster strongly together based on their common transcriptome-wide responses to different transfected sRNAs (*Figure 3B*), indicating the likely presence of batch effects (*Leek et al., 2010*) that could obscure detection of features associated with miRNA targeting.

A parameter known to confound the accurate measurement of mRNA responses on microarrays is the relative AU content within 3′ UTRs (*Elkon and Agami, 2008*). Indeed, when considering mRNAs without a canonical site to the transfected sRNA, we found that 3′-UTR AU content often correlated with mRNA fold changes. Moreover, the extent and direction of the correlation was similar for

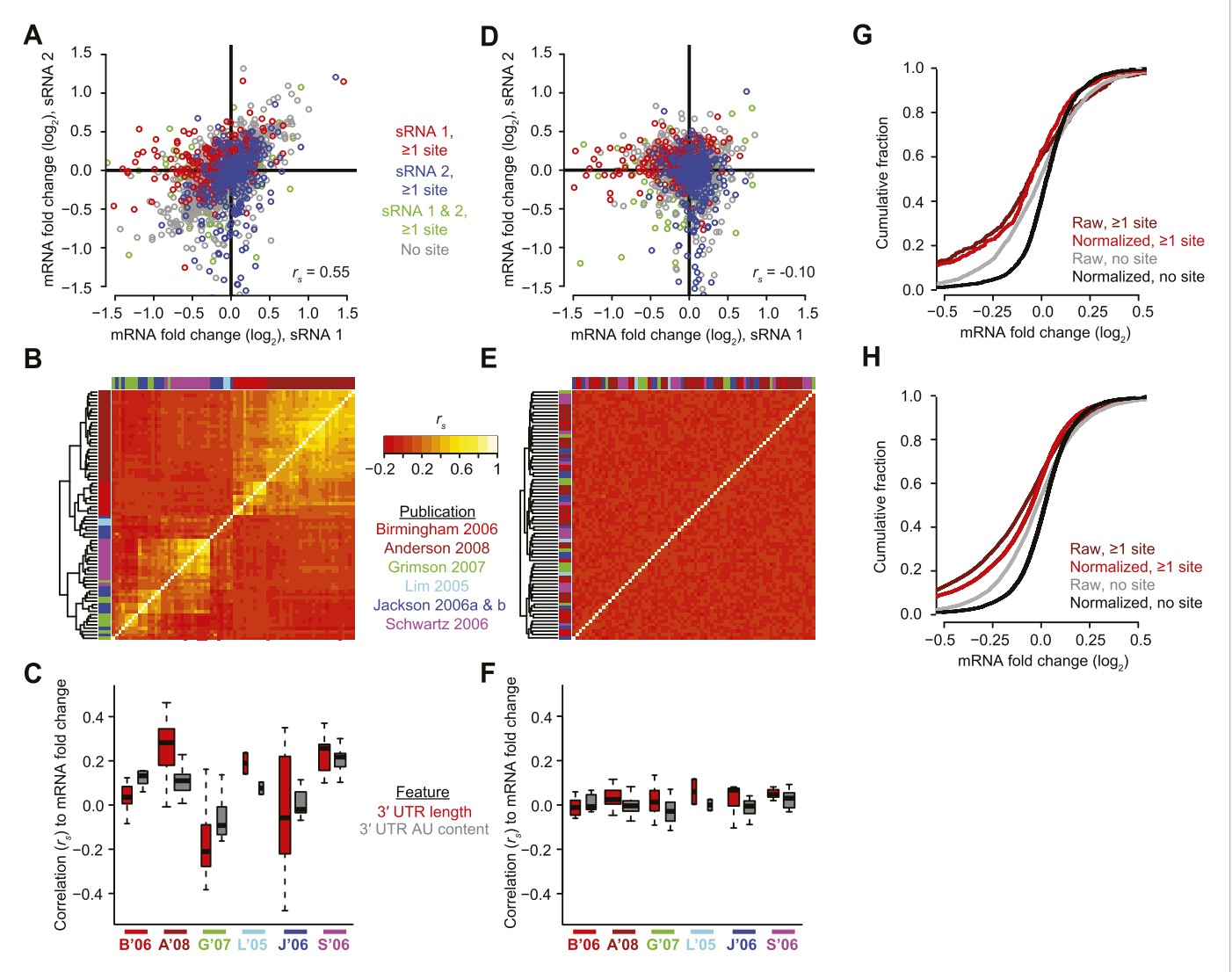

**Figure 3**. Pre-processing the microarray datasets to minimize nonspecific effects and technical biases. (**A**) Example of the correlated response of mRNAs after transfecting two unrelated sRNAs (sRNA 1 and 2, respectively). Results for mRNAs containing at least one canonical 7–8 nt 3'-UTR site for either sRNA 1, sRNA 2, or both sRNAs are highlighted in red, blue, and green, respectively. Values for mRNAs without such sites are in grey. All mRNAs were used to calculate the Spearman correlation ($r_s$). (**B**) Correlated responses observed in a compendium of 74 transfection experiments from six studies (colored as indicted in the publications list). For each pair of experiments, the $r_s$ value was calculated as in panel (**A**), colored as indicated in the key, and used for hierarchical clustering. (**C**) Study-dependent relationships between the responses of mRNAs to the transfected sRNA and either 3'-UTR length or 3'-UTR AU content, focusing on mRNAs without a canonical 7–8 nt 3'-UTR site to the sRNA. Boxplots indicate the median $r_s$ (bar), 25th and 75th percentiles (box), and the minimum of either 1.5 times the interquartile range or the most extreme data point (whiskers), with the width of the box proportional to the number of datasets used from each study. The studies are colored as in panel (**B**), abbreviating the first author and year. (**D**) Reduced correlation between the responses of mRNAs to unrelated sRNAs after applying the PLSR technique. This panel is as in (**A**) but plots the normalized mRNA fold changes. (**E**) Reduced correlations in results of the compendium experiments after applying the PLSR technique. This panel is as in (**B**) but plots the correlations after normalizing the mRNA fold changes. (**F**) Reduced study-dependent relationships between mRNA responses and either 3'-UTR length or 3'-UTR AU content. This panel is as in (**C**) but plots the correlations after normalizing the mRNA fold changes. (**G** and **H**) Cumulative distributions of fold changes for mRNAs containing at least one canonical 7–8 nt 3'-UTR site or no site either before normalization (raw) or after normalization (normalized). Panel (**G**) plots the results from experiments shown in (**A**) and (**D**), and (**H**) plots results from all 74 datasets.

The following figure supplement is available for figure 3:

**Figure supplement 1**. Reduced biases from derepression of endogenous miRNA targets.

different datasets from the same publication but differed when comparing to datasets from other publications (*Figure 3C*). A second parameter that helped explain the correlated sRNA-independent effects for related datasets was 3′-UTR length (*Saito and Satrom, 2012*), which exhibited patterns of correlation similar to those observed for 3′-UTR AU content (*Figure 3C*). Our observation that AU content and 3′-UTR length correlated so differently with global expression changes when comparing results from different publications helps explain why different 3′-UTR features previously seemed to have such variable predictive power in different experimental contexts (*Hausser et al., 2009*; *Wen et al., 2011*; *Gumienny and Zavolan, 2015*).

Another phenomenon known to systematically perturb the levels of mRNAs without sites to the transfected sRNA is the derepression of mRNAs with sites for endogenous miRNAs, presumably through competition between the transfected sRNA and the endogenous miRNAs for limiting components of the silencing pathway (*Khan et al., 2009*; *Saito and Satrom, 2012*). Statistically significant derepression was indeed observed for mRNAs with sites to eight of the 10 miRNA families most frequently sequenced in HeLa cells (*Figure 3—figure supplement 1A,B*).

To correct for biases that were independent of the sequence of the introduced sRNA, we used partial least-squares regression (PLSR) to estimate—for each transfection experiment—the component of the transcriptome response that was similar in other highly correlated experiments, and we then subtracted this estimate from the observed response (*Supplementary file 1*). Applying our technique to all the mRNAs in each of the 74 datasets largely eliminated the correlations observed between datasets (*Figure 3D–E*), as well as the correlations observed between mRNA fold changes and either AU content or 3′-UTR length (*Figure 3F*), which lowered the risk that these effects that are independent of the sRNA sequence would confound subsequent analyses of sRNA targeting efficacy. Moreover, our technique eliminated the signal for derepression of endogenous miRNA targets (*Figure 3—figure supplement 1C*), suggesting that it did the same for any other biases unrelated to the sequence of the transfected sRNA that have yet to be identified. Reducing these biases substantially reduced the variance in the response for mRNAs without sites to the sRNA, which substantially enhanced the net signal for sRNA-mediated repression of site-containing mRNAs observed in individual arrays (*Figure 3G*) and all arrays in aggregate (*Figure 3H*).

Previous studies of miRNA targeting have relied on 3′-UTR annotations from databases such as RefSeq, without accounting for abundant alternative 3′-UTR isoforms present in the tissue or cell line of interest (*Tian et al., 2005*). The presence of more than one abundant 3′-UTR isoform for a gene would confound interpretation of 3′-UTR-related features, such as 3′-UTR length, or distance from the closest 3′-UTR end (*Nam et al., 2014*). Moreover, the shorter 3′-UTR isoforms might not include some target sites, which would cause these sites to appear ineffective when in fact they are not present (*Sandberg et al., 2008*; *Mayr and Bartel, 2009*; *Lianoglou et al., 2013*; *Nam et al., 2014*). To avoid these complications, we examined 3′-UTR isoform quantifications previously generated for HeLa cells (*Nam et al., 2014*) using poly(A)-position profiling by sequencing (3P-seq) (*Jan et al., 2011*), and developed our model using the dominant mRNA from the subset of genes for which ≥90% of the 3P-seq tags corresponded to a single 3′-UTR isoform. To isolate the effects of single sites, we also used the subset of these mRNAs for which the 3′ UTR possessed a single seed match to the transfected sRNA (*Supplementary file 1*).

## Selecting features and building a regression model for target prediction

To improve our model of mammalian target-site efficacy, we considered 26 features as potentially informative of efficacy. These included features of the sRNAs, features of the sites (including their contexts and positions within the mRNAs), and features of the mRNAs, many of which had been used or at least considered in previous efforts (*Table 1*).

One of the 26 features was site $P_{CT}$ (probability of conserved targeting), which estimates the probability of the site being preferentially conserved because it is targeted by the cognate miRNA (*Friedman et al., 2009*). Prior to use, our $P_{CT}$ scores were updated to take advantage of improvements in both mouse and human 3′-UTR annotations (*Harrow et al., 2012*; *Flicek et al., 2014*), the additional sequenced vertebrate genomes aligned to the mouse and human genomes (*Karolchik et al., 2014*), and our expanded set of miRNA families broadly conserved among vertebrate species, which increased from 87 to 111 families (with the 111 including 16 isomiR families, that is, cases in which a second or third miRNA was produced from a pri-miRNA hairpin, through

**Table 1.** The 26 features considered in the models, highlighting the 14 robustly selected through stepwise regression (bold)

| Feature | Abbreviation | Description | Frequency chosen | | | |
|---|---|---|---|---|---|---|
| | | | 8mer | 7mer-m8 | 7mer-A1 | 6mer |
| **miRNA** | | | | | | |
| **3′-UTR target-site abundance** | **TA_3UTR** | **Number of sites in all annotated 3′ UTRs (*Arvey et al., 2010*; *Garcia et al., 2011*)** | **100%** | **100%** | **100%** | **100%** |
| ORF target-site abundance | TA_ORF | Number of sites in all annotated ORFs (*Garcia et al., 2011*) | 9.4% | 0.7% | 68.1% | 93.4% |
| **Predicted seed-pairing stability** | **SPS** | **Predicted thermodynamic stability of seed pairing (*Garcia et al., 2011*)** | **100%** | **100%** | **100%** | **100%** |
| **sRNA position 1** | **sRNA1** | **Identity of nucleotide at position 1 of the sRNA** | **68%** | **100%** | **99.7%** | **97.7%** |
| **sRNA position 8** | **sRNA8** | **Identity of nucleotide at position 8 of the sRNA** | **0%** | **0.8%** | **100%** | **100%** |
| **Site** | | | | | | |
| Site position 1 | site1 | Identity of nucleotide at position 1 of the site | N/A | 57.1% | N/A | 2% |
| **Site position 8** | **site8** | **Identity of nucleotide at position 8 of the site** | **0.8%** | **95.1%** | **99.4%** | **100%** |
| Site position 9 | site9 | Identity of nucleotide at position 9 of the site (*Lewis et al., 2005*; *Nielsen et al., 2007*) | 15.4% | 7.1% | 0.9% | 93.7% |
| Site position 10 | site10 | Identity of nucleotide at position 10 of the site (*Nielsen et al., 2007*) | 0.1% | 100% | 8.5% | 26.3% |
| **Local AU content** | **local_AU** | **AU content near the site (*Grimson et al., 2007*; *Nielsen et al., 2007*)** | **100%** | **100%** | **100%** | **100%** |
| **3′ supplementary pairing** | **3P_score** | **Supplementary pairing at the miRNA 3′ end (*Grimson et al., 2007*)** | **42.5%** | **100%** | **100%** | **100%** |
| Distance from stop codon | dist_stop | log$_{10}$(Distance of site from stop codon) | 62.4% | 10.8% | 8.7% | 25.7% |
| **Predicted structural accessibility** | **SA** | **log$_{10}$(Probability that a 14 nt segment centered on the match to sRNA positions 7 and 8 is unpaired)** | **100%** | **100%** | **100%** | **100%** |
| **Minimum distance** | **min_dist** | **log$_{10}$(Minimum distance of site from stop codon or polyadenylation site) (*Gaidatzis et al., 2007*; *Grimson et al., 2007*; *Majoros and Ohler, 2007*)** | **99.9%** | **100%** | **87.4%** | **100%** |
| **Probability of conserved targeting** | **$P_{CT}$** | **Probability of site conservation, controlling for dinucleotide evolution and site context (*Friedman et al., 2009*)** | **100%** | **100%** | **100%** | **20.8%** |
| **mRNA** | | | | | | |
| 5′-UTR length | len_5UTR | log$_{10}$(Length of the 5′ UTR) | 98.2% | 8.2% | 4.6% | 17.2% |
| **ORF length** | **len_ORF** | **log$_{10}$(Length of the ORF)** | **100%** | **100%** | **100%** | **100%** |
| **3′-UTR length** | **len_3UTR** | **log$_{10}$(Length of the 3′ UTR) (*Hausser et al., 2009*)** | **100%** | **100%** | **100%** | **100%** |
| 5′-UTR AU content | AU_5UTR | Fraction of AU nucleotides in the 5′ UTR | 13% | 38.9% | 91.1% | 31.3% |
| ORF AU content | AU_ORF | Fraction of AU nucleotides in the ORF | 1.2% | 72.4% | 28.4% | 35.8% |
| 3′-UTR AU content | AU_3UTR | Fraction of AU nucleotides in the 3′ UTR (*Robins and Press, 2005*; *Hausser et al., 2009*) | 5.4% | 73.3% | 65.3% | 80.6% |
| **3′-UTR offset-6mer sites** | **off6m** | **Number of offset-6mer sites in the 3′ UTR (*Friedman et al., 2009*)** | **65.9%** | **89.6%** | **99.8%** | **100%** |
| **ORF 8mer sites** | **ORF8m** | **Number of 8mer sites in the ORF (*Lewis et al., 2005*; *Reczko et al., 2012*)** | **99.5%** | **99.1%** | **100%** | **100%** |
| ORF 7mer-m8 sites | ORF7m8 | Number of 7mer-m8 sites in the ORF (*Reczko et al., 2012*) | 4.7% | 4.3% | 85.3% | 100% |
| ORF 7mer-A1 sites | ORF7A1 | Number of 7mer-A1 sites in the ORF (*Reczko et al., 2012*) | 68.4% | 34.2% | 97.8% | 98.4% |
| ORF 6mer sites | ORF6m | Number of 6mer sites in the ORF (*Reczko et al., 2012*) | 91% | 13.3% | 0.7% | 36.7% |

The feature description does not include the scaling performed (**Table 3**) to generate more comparable regression coefficients.

either conserved expression of miRNAs from both arms of the hairpin or conserved 5′ heterogeneity). Using these updates increased sensitivity, with our estimate for the number of human 3′-UTR sites conserved above background increasing from ~46,400 (*Friedman et al., 2009*) to ~62,300. The $P_{CT}$ score on its own correlates with site efficacy, and when using the same set of 3′ UTRs this correlation increased only modestly for the new scores (data not shown), consistent with the notion that the evolutionary signal was already nearly saturated in the previous analysis of 23 species spanning the vertebrate tetrapods (*Friedman et al., 2009*). Nonetheless, we used our updated $P_{CT}$ score as a feature for sites of broadly conserved miRNAs within our training set.

A second feature that we re-evaluated was the predicted structural accessibility of the site. As scored previously, the degree to which the site nucleotides were predicted to be free of pairing to flanking 3′-UTR regions was not informative after controlling for the contribution of local AU content (*Grimson et al., 2007*). However, analysis inspired by work on siRNA site accessibility (*Tafer et al., 2008*) suggested an improved scoring scheme for this feature. For this analysis we used RNAplfold (*Bernhart et al., 2006*) to predict the unpaired probabilities for variable-sized windows in the proximity of the site and then examined the relationship between these probabilities and the repression associated with sites in our compendium of normalized datasets, while controlling for local AU content and other features of the context+ model (*Figure 4A*). Based on these results, which resembled those reported previously (*Tafer et al., 2008*), we scored predicted structural accessibility (SA) as proportional to the $\log_{10}$ value of the unpaired probability for a 14-nt region centered on the match to miRNA nucleotides 7 and 8.

Having assembled a set of candidate features, we used the *stepAIC* function from the 'MASS' R package (*Venables and Ripley, 2002*) to determine which features were most useful for modeling site efficacy. This function uses stepwise regression to build models with increasing numbers of features until it reaches the optimal Akaike Information Criterion (AIC) value. The AIC evaluates the tradeoff between the benefit of increasing the likelihood of the regression fit and the cost of increasing the complexity of the model by adding more variables. For each of the four seed-matched site types, models were built for 1000 samples of the dataset. Each sample included 70% of the mRNAs with single sites to the transfected sRNA from each experiment (randomly selected without replacement), reserving the remaining 30% as a test set. Compared to our context-only and context+ models (*Grimson et al., 2007*; *Garcia et al., 2011*), the new stepwise regression models were significantly better at predicting site efficacy when evaluated using their corresponding held-out test sets, as illustrated for the each of four site types (*Figure 4B*).

Reasoning that features most predictive would be robustly selected, we focused on 14 features selected in nearly all 1000 bootstrap samples for at least two site types (*Table 1*). These included all three features considered in our original context-only model (minimum distance from 3′-UTR ends, local AU composition and 3′-supplementary pairing), the two added in our context+ model (SPS and TA), as well as nine additional features (3′-UTR length, ORF length, predicted SA, the number of offset-6mer sites in the 3′ UTR and 8mer sites in the ORF, the nucleotide identity of position 8 of the target, the nucleotide identity of positions 1 and 8 of the sRNA, and site conservation). Other features were frequently selected for only one site type (e.g., ORF 7mer-A1 sites, ORF 7mer-m8 sites, and 5′-UTR length; *Table 1*). Presumably these and other features were not robustly selected because either their correlation with targeting efficacy was very weak (e.g., the 7 nt ORF sites) or they were strongly correlated to a more informative feature, such that they provided little additional value beyond that of the more informative feature (e.g., 3′-UTR AU content compared to the more informative feature, local AU content).

Using the 14 robustly selected features, we trained multiple linear regression models on all of the data. The resulting models, one for each of the four site types, were collectively called the context++ model (*Figure 4C* and *Figure 4—source data 1*). For each feature, the sign of the coefficient indicated the nature of the relationship. For example, mRNAs with either longer ORFs or longer 3′ UTRs tended to be more resistant to repression (indicated by a positive coefficient), whereas mRNAs with either structurally accessible target sites or ORF 8mer sites tended to be more prone to repression (indicated by a negative coefficient). Based on the relative magnitudes of the regression coefficients, some newly incorporated features, such as 3′-UTR length, ORF length, and SA, contributed similarly to features previously incorporated in the context+ model, such as SPS, TA, and local AU (*Figure 4C*). New features with an intermediate level of influence included the number of ORF 8mer sites and site conservation as well as the presence of a 5′ G in the sRNA (*Figure 4C*), the

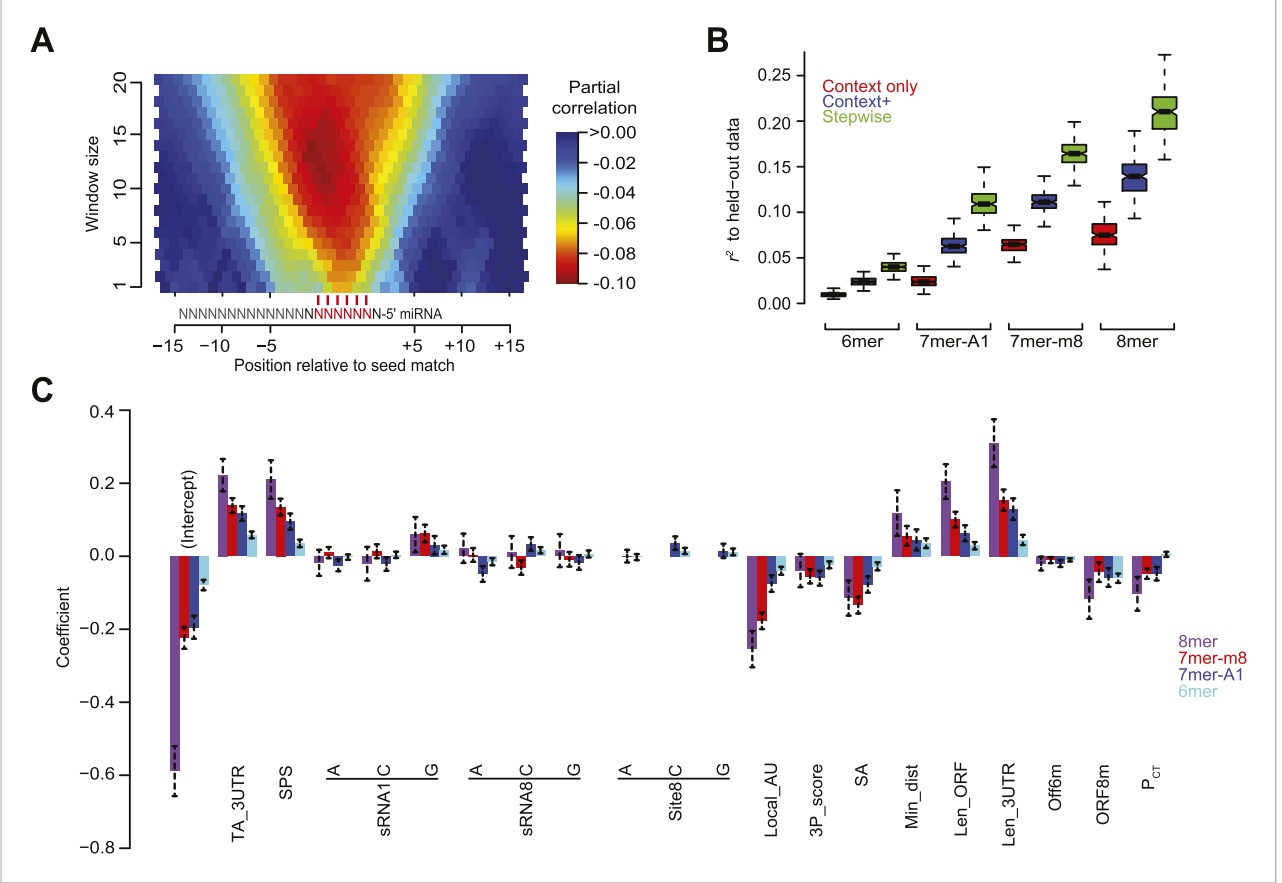

**Figure 4**. Developing a regression model to predict miRNA targeting efficacy. (**A**) Optimizing the scoring of predicted structural accessibility. Predicted RNA structural accessibility scores were computed for variable-length windows within the region centered on each canonical 7–8 nt 3′-UTR site. The heatmap displays the partial correlations between these values and the repression associated with the corresponding sites, determined while controlling for local AU content and other features of the context+ model (*Garcia et al., 2011*). (**B**) Performance of the models generated using stepwise regression compared to that of either the context-only or context+ models. Shown are boxplots of $r^2$ values for each of the models across all 1000 sampled test sets, for mRNAs possessing a single site of the indicated type. For each site type, all groups significantly differ ($P < 10^{-15}$, paired Wilcoxon sign-rank test). Boxplots are as in *Figure 3C*. (**C**) The contributions of site type and each of the 14 features of the context++ model. For each site type, the coefficients for the multiple linear regression are plotted for each feature. Because features are each scored on a similar scale, the relative contribution of each feature in discriminating between more or less effective sites is roughly proportional to the absolute value of its coefficient. Also plotted are the intercepts, which roughly indicate the discriminatory power of site type. Dashed bars indicate the 95% confidence intervals of each coefficient.

The following source data is available for figure 4:

**Source data 1**. Coefficients of the trained context++ model corresponding to each site type.

latter perhaps a consequence of differential sRNA loading efficiency. The weakest features included the sRNA and target position 8 identities as well as the number of offset-6mer sites. The identity of sRNA nucleotide 8 exhibited a complex pattern that was site-type dependent. Relative to a position-8 U in the sRNA, a position-8 C further decreased efficacy of sites with a mismatch at this position (6mer or 7mer-A1 sites), whereas a position-8 A had the opposite effect (*Figure 4C*). Similarly, a position-8 C in the site also conferred decreased efficacy of 6mer and 7mer-A1 sites relative to a position-8 U in the site (*Figure 4C*). Allowing interaction terms when developing the model, including a term that captured the potential interplay between these positions, did not provide sufficient benefit to justify the more complex model.

## Improvement over previous methods

We compared the predictive performance of our context++ model to that of the most recent versions of 17 in silico tools for predicting miRNA targets, including AnTar (*Wen et al., 2011*), DIANA-microT-

CDS (*Reczko et al., 2012*), ElMMo (*Gaidatzis et al., 2007*), MBSTAR (*Bandyopadhyay et al., 2015*), miRanda-MicroCosm (*Griffiths-Jones et al., 2008*), miRmap (*Vejnar and Zdobnov, 2012*), mirSVR (*Betel et al., 2010*), miRTarget2 (*Wang and El Naqa, 2008*), MIRZA-G (*Gumienny and Zavolan, 2015*), PACCMIT-CDS (*Marin et al., 2013*), PicTar2 implemented for predictions conserved through mammals, chicken, or fish (PicTarM, PicTarC, and PicTarF, respectively) (*Anders et al., 2012*), PITA (*Kertesz et al., 2007*), RNA22 (*Miranda et al., 2006*), SVMicrO (*Liu et al., 2010*), TargetRank (*Nielsen et al., 2007*), and TargetSpy (*Sturm et al., 2010*); as well as successive versions of TargetScan, which offer context scores (*Grimson et al., 2007*), $P_{CT}$ scores (*Friedman et al., 2009*), or context+ scores (*Garcia et al., 2011*) as options for ranking predictions (TargetScan5, TargetScan.$P_{CT}$, or TargetScan6, respectively) for either all mRNAs with a canonical 7–8 nt 3′-UTR site (TargetScan.All) or those with only broadly conserved sites (TargetScan.Cons). To the best of our knowledge, algorithms excluded from the comparison either were not de novo prediction algorithms (relying on consensus techniques or experimental data), did not provide a pre-computed database of results, or lacked a numerical value (or ranking) of either target-prediction confidence or mRNA responsiveness. To test the performance of the included methods, we used the results of seven microarray datasets that each monitor mRNA changes after transfection of a conserved miRNA into HCT116 cells containing a hypomorphic mutant for Dicer (*Linsley et al., 2007*). These datasets differ from those used during development and training of our model with respect to both the cell type and the identities of the sRNAs. To prevent our model from gaining an advantage over methods that used standard 3′-UTR annotations, we used RefSeq-annotated 3′ UTRs (rather than 3P-seq–supported annotations) to generate the context++ test-set predictions. For genes with multiple annotated 3′ UTRs we chose the longest isoform because the microarray probes of the test set often matched only this isoform. For each 3′ UTR containing multiple sites to the cognate miRNA, the context++ scores of individual sites were summed to generate the total context++ score to be used to rank that predicted target.

The number of potential miRNA–mRNA interactions considered by the different methods varied greatly (*Figure 5A*), which reflected the varied strategies and priorities of these prediction efforts. Out of a concern for prediction specificity, many efforts only consider interactions involving 7–8 nt seed-matched sites. Accordingly, we first tested how well each of the methods predicted the repression of mRNAs with at least one canonical 7–8 nt 3′-UTR site (*Figure 5B*). The context++ model performed substantially better than the most predictive published model, which was TargetScan6.All. Of algorithms derived from other groups, DIANA-microT-CDS, miRTarget2, miRanda-miRSVR, MIRZA-G (and its derivatives), and TargetRank were the most predictive, with performance within range of TargetScan5.All (*Figure 5B*).

Part of the reason that some algorithms performed more poorly is that they consider relatively few potential miRNA–target interactions (*Figure 5A*). For example, the drop in performance observed between TargetScan.All and TargetScan.Cons illustrates the effect of limiting analysis to the more highly conserved sites. Nonetheless, the performance of TargetScan.Cons relative to other methods that consider relatively few sites shows that a signal can be observed in this assay even when a very limited number of interactions are scored (*Figure 5A,B*), presumably because much of the functional targeting is through conserved interactions. Indeed, the performance of ElMMO and TargetScan.$P_{CT}$ illustrate what can be achieved by scoring just the extent of site conservation and no other parameter.

In an attempt to maximize prediction sensitivity, some efforts consider many interactions that lack a canonical 7–8 nt 3′-UTR site (*Figure 5A*). However, all of these algorithms performed poorly in predicting the response of mRNAs lacking such sites (*Figure 5C*). The two algorithms achieving any semblance of prediction accuracy did so by predicting some of the canonical interactions with known marginal efficacy. These were DIANA-microT-CDS, which captured modest effects of canonical sites in ORFs (*Reczko et al., 2012*; *Marin et al., 2013*), and the context++ model, which captured the modest effects of canonical 6mers in 3′ UTRs (as modified by the 14 features, which included offset 6mers and 8mer ORF sites) (*Figure 5C*). The algorithms designed to identify many non-canonical sites performed much more poorly in this test ($r^2 < 0.004$), consistent with the idea that the vast majority of mRNAs without canonical sites either do not change in response to the miRNA or change in an unpredictable fashion as a secondary effect of introducing the miRNA.

Another way to evaluate the performance of targeting algorithms is to examine the repression of the top predicted targets. Compared to the $r^2$ test, this approach does not penalize efforts that either impose more stringent cutoffs to achieve higher prediction specificity or implement scoring schemes that are not designed to correlate directly with site efficacy. Perhaps most importantly, this approach

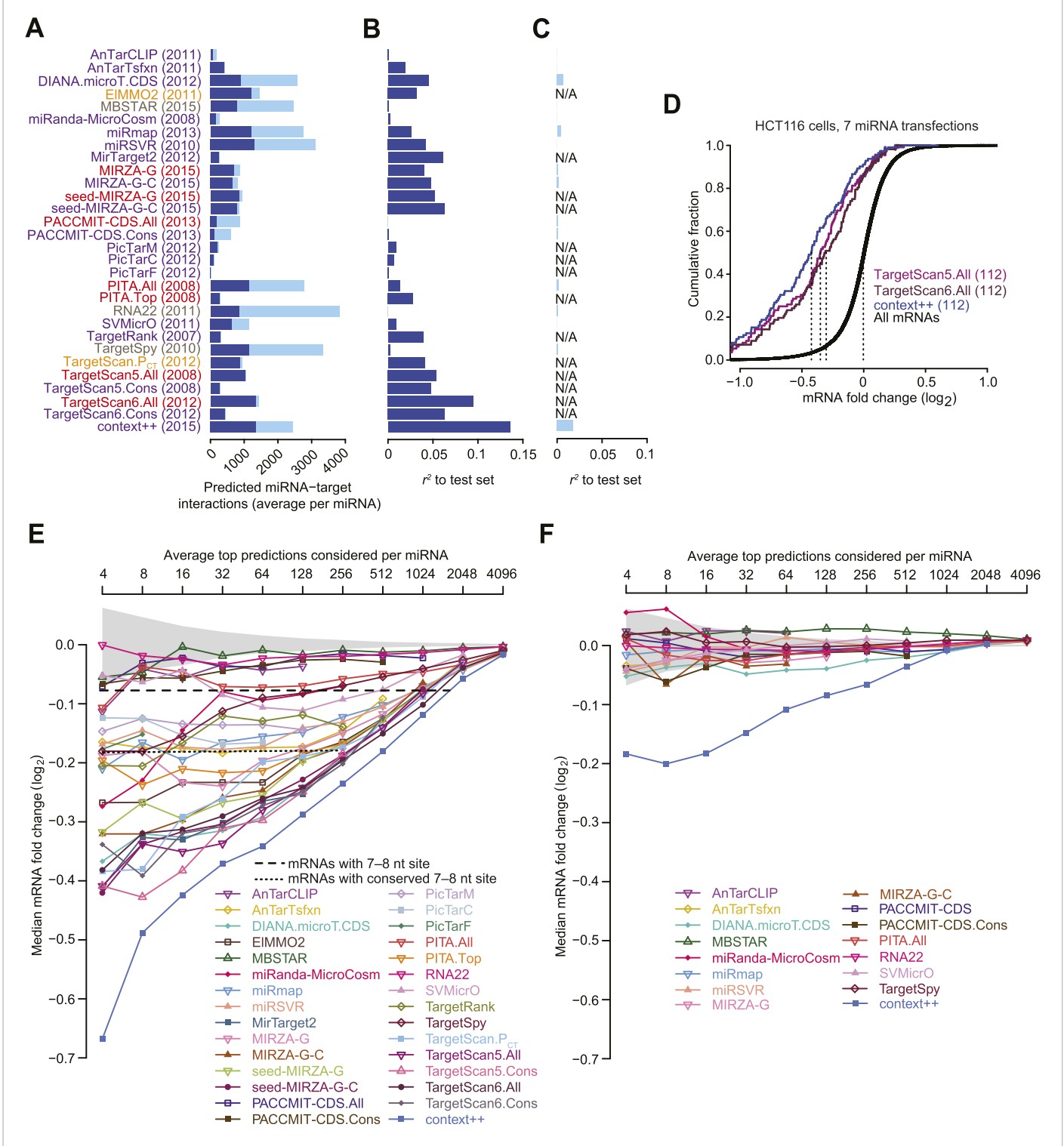

**Figure 5**. Performance of target prediction algorithms on a test set of seven experiments in which miRNAs were individually transfected into HCT116 cells. (**A**) Average number of targets predicted by the indicated algorithm for each of the seven miRNAs in the test set (let-7c, miR-16, miR-103, miR-106b, miR-200b, miR-200a, and miR-215). The numbers of predictions with at least one canonical 7–8 nt 3′-UTR site to the transfected miRNA (dark blue) are distinguished from the remaining predictions (light blue). Names of algorithms are colored according to whether they consider only sequence or thermodynamic features of site pairing (grey), only site conservation (orange), pairing and contextual features of a site (red), or pairing, contextual features, and site conservation (purple). The most recently updated predictions were downloaded, with year that those predictions were released indicated in

*Figure 5. continued on next page*

*Figure 5. Continued*

parentheses. (**B** and **C**) Extent to which the predictions explain the mRNA fold changes observed in the test set. For predictions tallied in panel (**A**), the explanatory power, as evaluated by the $r^2$ value for the relationship between the scores of the predictions and the observed mRNA fold changes in the test set, is plotted for either mRNAs with 3′ UTRs containing at least one canonical 7–8 nt 3′-UTR site (**B**) or other mRNAs (**C**). Algorithms designed to evaluate only targets with seed-matched 7–8 nt 3′-UTR sites are labeled 'N/A' in (**C**). (**D**) Repression of the top predictions of the context++ model and of our previous two models, focusing on an average of 16 top predicted targets per miRNA in the test set. The dotted lines indicate the median fold-change value for each distribution, otherwise as in *Figure 1A*. (**E** and **F**) Median mRNA fold changes observed in the test set for top-ranked predicted targets, considering either all predictions (**E**) or only those with 3′ UTRs lacking at least one canonical 7–8 nt site (**F**). For each algorithm listed in panel (**A**), all reported predictions for the seven miRNAs were ranked according to their scores, and the indicated sliding threshold of top predictions was implemented. For example, at the threshold of 4, the 28 predictions with the top scores were identified (an average of 4 predictions per miRNA, allowing miRNAs with more top scores to contribute more predictions), mRNA fold-change values from the cognate transfections were collected, and the median value was plotted. When the threshold exceeded the number of reported predictions, no value was plotted. Also plotted is the median mRNA fold change for all mRNAs with at least one cognate canonical 7–8 nt site in their 3′ UTR (dashed line; an average of 1366 mRNAs per miRNA), the median fold change for all mRNAs with at least one conserved cognate canonical 7–8 nt site in their 3′ UTR (dotted line; an average of 461 mRNAs per miRNA), and the 95% interval for the median fold change of randomly selected mRNAs, determined using 1000 resamplings (without replacement) at each cutoff (shading). Conserved sites were defined as in TargetScan6, with conservation cutoffs for each site type set at different branch-length scores (cutoffs of 0.8, 1.3, and 1.6 for 8mer, 7mer-m8, and 7mer-A1 sites, respectively).

The following figure supplement is available for figure 5:

**Figure supplement 1**. Performance of miRNA prediction algorithms on the test set.

aligns with the goals of a biologist considering the top-ranked predictions in an attempt to focus on those most likely to undergo substantial repression. When choosing an average of 16 predicted targets for each of the seven test-set miRNAs, we found that these top 112 predictions of the context++ model were significantly more repressed than the top predictions from earlier versions of TargetScan (*Figure 5D*) and the top predictions of the other algorithms (*Figure 5—figure supplement 1A*).

Despite the success of the context++ model, not all of the fold changes for its top predicted targets were negative; for the test set, the distribution of these fold changes intersected 0.0 at a cumulative fraction of 0.92, indicating that mRNAs for 8% of the top predictions increased rather than decreased with transfection of the cognate miRNA (*Figure 5D*). In principle, these mRNAs could still be authentic targets that are repressed in these cells but nonetheless had increased expression values because either experimental noise or secondary effects of introducing the miRNA overwhelmed the signal for miRNA-mediated repression. Alternatively, some or all of these mRNAs could be false-positive predictions. Because only half of the false-positive predictions would be expected to have positive fold changes in the presence of the miRNA, our best estimate of the upper limit on the false-positive predictions was 2 × 8%, or 16%, at this cutoff (for which an average of 16 top predictions per miRNA is considered). At the same cutoff, the distribution of fold changes for each of the previous algorithms intersected 0.0 at a cumulative fractions ranging from 0.50–0.88 (*Figure 5—figure supplement 1A*), which implied lower prediction specificity than that observed for the context++ model, with correspondingly higher estimates for the upper limits of false positives among their top predictions, ranging from 24–100%.

To evaluate the performance of top-ranked predictions more systematically, we examined median repression of the predicted targets over a broad spectrum of cutoffs, ranging from an average of 4–4096 predictions per miRNA (*Figure 5E*). Regardless of the cutoff, the top context++ predictions were the most repressed. The top predictions of most other algorithms were repressed significantly more than expected by chance, although the median repression of some (MBSTAR, RNA22, PACCMIT-CDS, and AnTarCLIP) did not exceed the median repression of all mRNAs with a canonical 7–8 nt 3′-UTR site (*Figure 5E*). Plotting average fold changes rather than median fold changes resulted in very similar relative performances (*Figure 5—figure supplement 1B*).

After eliminating interactions that could involve canonical 7–8 nt 3′-UTR sites, the remaining top predictions were modestly repressed at best (*Figure 5F* and *Figure 5—figure supplement 1C*). The most repressed predicted targets without canonical 7–8 nt 3′-UTR sites were those of the context++ model, which scored predictions with canonical 6mer 3′-UTR sites. For algorithms designed to identify many non-canonical sites, the top predictions without 7–8 nt 3′-UTR sites were essentially

unresponsive to the transfected miRNA, which indicated that if effective non-canonical sites for these seven miRNAs exist, they are not enriched among the top predictions of these algorithms.

## Similar response of targets predicted from the model and the most informative CLIP experiments

We used our context++ model to overhaul the TargetScan predictions (as described in the next section), and as a third way of testing this model, we compared the performance of these TargetScan7 predictions with that of in vivo CLIP experiments. When doing this comparison we took care to evaluate sets of predictions that each were the same size as the cognate set of CLIP-supported targets, whereas some previous analyses compare expansive sets of computational predictions (e.g., all mRNAs with a 6mer site) to relatively small sets of biochemically supported predictions (*Chi et al., 2009*; *Lipchina et al., 2011*; *Loeb et al., 2012*; *Grosswendt et al., 2014*; *Tan et al., 2014*). mRNAs with expression signals approaching the array background were not considered. This exclusion was particularly important when comparing to CLIP results; CLIP can only evaluate mRNAs expressed in the cells, which would impart a trivial relative advantage if the computational predictions included targets that appeared unresponsive because they were expressed below the array background. The non-canonical CLIP-supported targets were also not considered, as we had already shown that they do not respond to the miRNA (*Figure 1* and *Figure 1—figure supplements 1–4*), and we did not want the inclusion of these easily recognized false positives to impart a disadvantage to CLIP. Regardless of the set of canonical CLIP-supported targets examined, we did not find a setting in which they responded significantly better than did the cohort of TargetScan7 predictions, and in some cases, the TargetScan7 predictions performed significantly better (*Figure 6A–J*). Similar results were observed when comparing the repression of our predictions to that of mRNAs identified biochemically without crosslinking, using either pulldown-seq or IMPACT-seq (*Tan et al., 2014*), again focusing on only mRNAs with canonical sites (*Figure 6K,L*). Thus, for identifying consequential miRNA–target interactions, the TargetScan7 model is not only more convenient than experimental determination of binding sites, it is also at least as effective. The analogous conclusion was reached from analyses that used the context++ model without using the improved annotation and quantification of 3′-UTR isoforms (data not shown).

As mentioned earlier, mRNAs that increase rather than decrease in the presence of the miRNA can indicate the presence of false positives in a set of candidate targets. Examination of the mRNA fold-change distributions from the perspective of false positives revealed no advantage of the experimental approaches over our predictions. When compared to the less informative CLIP datasets, the TargetScan7 predictions included fewer mRNAs that increased, and when compared to the CLIP datasets that performed as well as the predictions, the TargetScan7 predictions included a comparable number of mRNAs that increased, implying that the TargetScan7 predictions had no more false-positive predictions than did the best experimental datasets.

Because some sets of canonical biochemically supported targets performed as well as their cohort of top TargetScan7 predictions, we considered the utility of focusing on mRNAs identified by both approaches. In each comparison, the set of mRNAs that were both canonical biochemically supported targets and within the cohort of top TargetScan7 predictions tended to be more responsive. However, these intersecting subsets included much fewer mRNAs than the original sets, and when compared to an equivalent number of top TargetScan7 predictions, each intersecting set performed no better than did its cohort of top TargetScan7 predictions (*Figure 6*). Therefore, considering the CLIP results to restrict the top predictions to a higher-confidence set is useful but not more useful than simply implementing a more stringent computational cutoff. Likewise, taking the union of the CLIP-supported targets and the cohort of predictions, rather than the intersection, did not generate a set of targets that was more responsive than an equivalent number of top TargetScan7 predictions (data not shown).

## The TargetScan database (v7.0)

As already mentioned, we used the context++ model to rank miRNA target predictions to be presented in version 7 of the TargetScan database (targetscan.org), thereby making our results accessible to others working on miRNAs. For simplicity, we had developed the context++ model using mRNAs without abundant alternative 3′-UTR isoforms, and to make fair comparisons with the

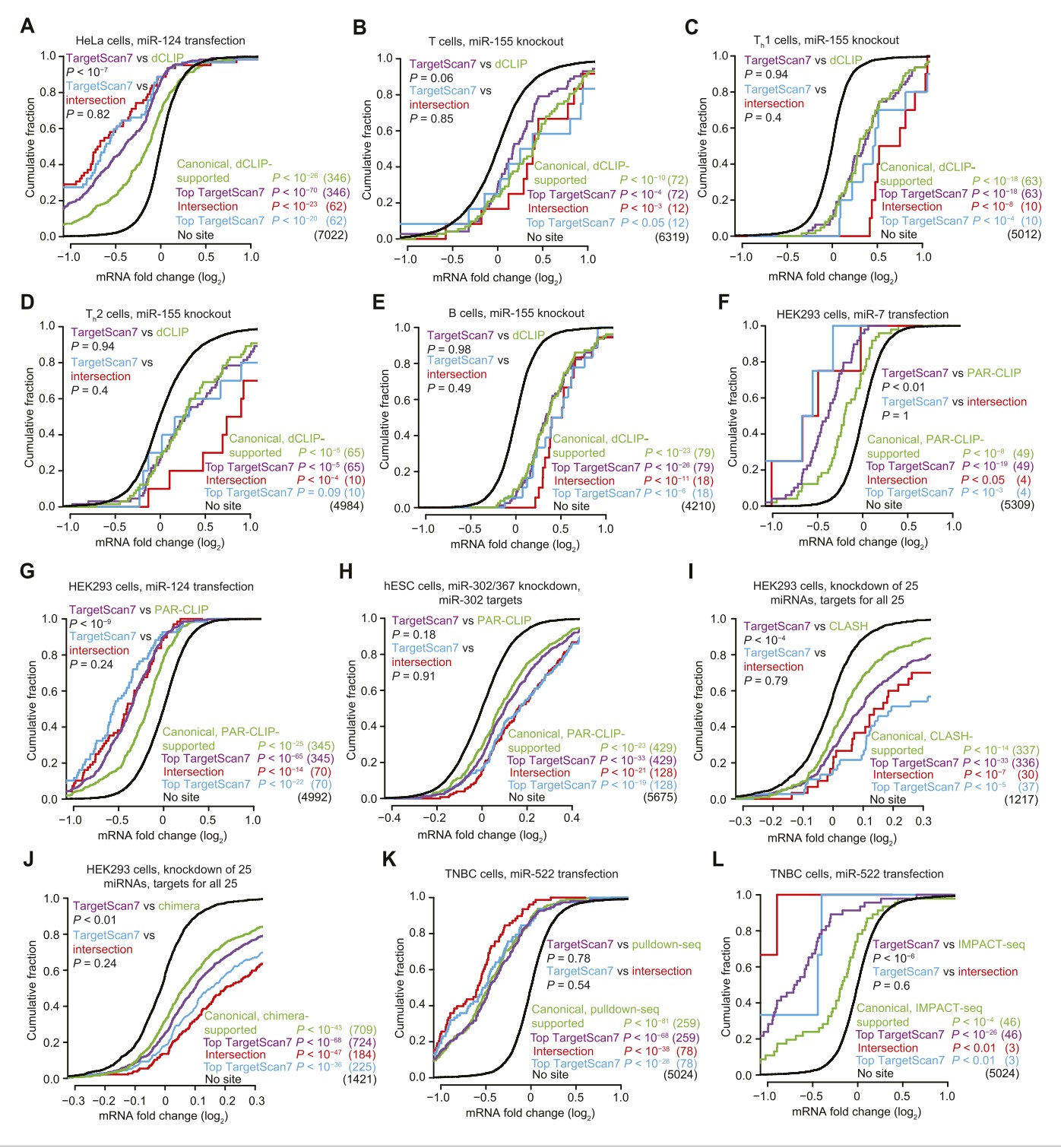

**Figure 6**. Response of predictions and mRNAs with experimentally supported canonical binding sites. (**A–E**) Comparison of the top TargetScan7 predicted targets to mRNAs with canonical sites identified from dCLIP in either HeLa cells with and without transfected miR-124 (*Chi et al., 2009*) or lymphocytes with and without miR-155 (*Loeb et al., 2012*). Plotted are cumulative distributions of mRNA fold changes after transfection of miR-124 in HeLa cells (**A**), or after genetic ablation of *miR-155* in either T cells (**B**), T$_h$1 cells (**C**), T$_h$2 cells (**D**), and B cells (**E**) (one-sided K–S test, *P* values). For genes with alternative last exons, the analysis considered the score of the most abundant alternative last exon, as assessed by 3P-seq tags (as is the default for TargetScan7 when ranking predictions). Each dCLIP-identified mRNA was required to have a 3'-UTR CLIP cluster with at least one canonical site to the cognate miRNA (including 6mers but not offset 6mers). Each intersection mRNA (red) was found in both the dCLIP set and top TargetScan7 set. Similarity

*Figure 6. continued on next page*

*Figure 6. Continued*

between performance of the TargetScan7 and dCLIP sets (purple and green, respectively) and TargetScan7 and intersection sets (blue and red, respectively) was tested (two-sided K–S test, *P* values); the number of mRNAs analyzed in each category is in parentheses. TargetScan7 scores for mouse mRNAs were generated using human parameters for all features. (**F**–**H**) Comparison of top TargetScan7 predicted targets to mRNAs with canonical binding sites identified using photoactivatable-ribonucleoside-enhanced CLIP (PAR-CLIP) (*Hafner et al., 2010*; *Lipchina et al., 2011*). Plotted are cumulative distributions of mRNA fold changes after either transfecting miR-7 (**F**) or miR-124 (**G**) into HEK293 cells, or knocking down miR-302/367 in hESCs (**H**). Otherwise these panels are as in (**A**–**E**). (**I**) Comparison of top TargetScan7 predicted targets to mRNAs with canonical sites identified using CLASH (*Helwak et al., 2013*). Plotted are cumulative distributions of mRNA fold changes after knockdown of 25 miRNAs from 14 miRNA families in HEK293 cells. For each of these miRNA families, a cohort of top TargetScan7 predictions was chosen to match the number of mRNAs with CLASH-identified canonical sites, and the union of these TargetScan7 cohorts was analyzed. The total number of TargetScan7 predictions did not match the number of CLASH-identified targets due to slightly different overlap between mRNAs targeted by different miRNAs. Otherwise these panels are as in (**A**–**E**). (**J**) Comparison of top TargetScan7 predicted targets to mRNAs with chimera-identified canonical sites (*Grosswendt et al., 2014*). Otherwise this panel is as in (**I**). (**K**) Comparison of top TargetScan7 predicted targets to mRNAs with canonical binding sites within 3′ UTRs of mRNAs identified using pulldown-seq (*Tan et al., 2014*). Plotted are cumulative distributions of mRNA fold changes after transfecting miR-522 into triple-negative breast cancer (TNBC) cells. Otherwise this panel is as in (**A**–**E**). (**L**) Comparison of top TargetScan7 predicted targets to mRNAs with canonical sites identified using IMPACT-seq (*Tan et al., 2014*). Otherwise this panel is as in (**K**).

output of previous models, we had tested the context++ model using only the longest RefSeq-annotated isoform. Nevertheless, considering the usage of alternative 3′-UTR isoforms, which can influence both the presence and scoring of target sites, significantly improves the performance of miRNA targeting models (*Nam et al., 2014*). Thus, our overhaul of the TargetScan predictions incorporated both the context++ scores and current isoform information when ranking mRNAs with canonical 7–8 nt miRNA sites in their 3′ UTRs. The resulting improvements applied to the predictions centered on human, mouse, and zebrafish 3′ UTRs (TargetScanHuman, TargetScanMouse, and TargetScanFish, respectively); and by 3′-UTR homology, to the conserved and nonconserved predictions in chimp, rhesus, rat, cow, dog, opossum, chicken, and frog; as well as to the conserved predictions in 74 other sequenced vertebrate species, thereby providing a valuable resource for placing miRNAs into gene-regulatory networks.

Because the main gene-annotation databases (e.g., RefSeq and Ensembl/Gencode) are still in the process of incorporating the information available on 3′-UTR isoforms, the first step in the TargetScan overhaul was to compile a set of reference 3′ UTRs that represented the longest 3′-UTR isoforms for representative ORFs of human, mouse, and zebrafish. These representative ORFs were chosen among the set of transcript annotations sharing the same stop codon, with alternative last exons generating multiple representative ORFs per gene. The human and mouse databases started with Gencode annotations (*Harrow et al., 2012*), for which 3′ UTRs were extended, when possible, using RefSeq annotations (*Pruitt et al., 2012*), recently identified long 3′-UTR isoforms (*Miura et al., 2013*), and 3P-seq clusters marking more distal cleavage and polyadenylation sites (*Nam et al., 2014*). Zebrafish reference 3′ UTRs were similarly derived in a recent 3P-seq study (*Ulitsky et al., 2012*).

For each of these reference 3′-UTR isoforms, 3P-seq datasets were used to quantify the relative abundance of tandem isoforms, thereby generating the isoform profiles needed to score features that vary with 3′-UTR length (len_3UTR, min_dist, and off6m) and assign a weight to the context++ score of each site, which accounted for the fraction of 3′-UTR molecules containing the site (*Nam et al., 2014*). For each representative ORF, our new web interface depicts the 3′-UTR isoform profile and indicates how the isoforms differ from the longest Gencode annotation (*Figure 7*).

3P-seq data were available for seven developmental stages or tissues of zebrafish, enabling isoform profiles to be generated and predictions to be tailored for each of these. For human and mouse, however, 3P-seq data were available for only a small fraction of tissues/cell types that might be most relevant for end users, and thus results from all 3P-seq datasets available for each species were combined to generate a meta 3′-UTR isoform profile for each representative ORF. Although this approach reduces accuracy of predictions involving differentially expressed tandem isoforms, it nonetheless outperforms the previous approach of not considering isoform abundance at all, presumably because isoform profiles for many genes are highly correlated in diverse cell types (*Nam et al., 2014*).

For each 6–8mer site, we used the corresponding 3′-UTR profile to compute the context++ score and to weight this score based on the relative abundance of tandem 3′-UTR isoforms that contained

the site (*Nam et al., 2014*). Scores for the same miRNA family were also combined to generate cumulative weighted context++ scores for the 3′-UTR profile of each representative ORF, which provided the default approach for ranking targets with at least one 7–8 nt site to that miRNA family. Effective non-canonical site types, that is, 3′-compensatory and centered sites, were also predicted. Using either the human or mouse as a reference, predictions were also made for orthologous 3′ UTRs of other vertebrate species.

As an option for tetrapod species, the user can request that predicted targets of broadly conserved miRNAs be ranked based on their aggregate $P_{CT}$ scores (*Friedman et al., 2009*), as updated in this study. The user can also obtain predictions from the perspective of each protein-coding gene, viewed either as a table of miRNAs (ranked by either cumulative weighted context++ score or aggregate $P_{CT}$ score) or as the mapping of 7–8 nt sites (as well as non-canonical sites) shown beneath the 3′-UTR profile and above the 3′-UTR sequence alignment (*Figure 7*). A flowchart summarizing the TargetScan overhaul is provided (*Figure 7—figure supplement 1*).

## Discussion

Starting with an expanded and improved compendium of sRNA transfection datasets, we identified 14 features that each correlate with target repression and add predictive value when incorporated into a quantitative model of miRNA targeting efficacy. This model performed better than previous models and at least as well as the best high-throughput CLIP approaches.

Because our model was trained on data derived from a single cell type, a potential concern was its generalizability to other cell types. Heightening this concern is the recent report of widespread dependency of miRNA-mediated repression on cellular context (*Erhard et al., 2014*). However, other work addressing this question shows that after accounting for the different cellular repertoires of expressed mRNAs, the target response is remarkably consistent between different cell types, with alternative usage of 3′-UTR isoforms being the predominant mechanism shaping cell-type-specific differences in miRNA targeting (*Nam et al., 2014*). Testing the model across diverse cell types confirmed its generalizability; it performed at least as well as the best high-throughput CLIP approaches in each of the contexts examined (*Figure 6*). Of course, this testing was restricted to only those predicted targets that were expressed in each cellular context. Likewise, to achieve this highest level of performance, any future use of our model or its predictions would also require filtering of the predictions to focus on only the miRNAs and mRNAs co-expressed in the cells of interest.

One of the more interesting features incorporated into the context++ model is SA (the predicted structural accessibility of the site). Freedom from occlusive mRNA structure has long been considered a site-efficacy determinant (*Robins et al., 2005*; *Ameres et al., 2007*; *Kertesz et al., 2007*; *Long et al., 2007*; *Tafer et al., 2008*) and proposed as the underlying mechanistic explanation for the utility of other features, including global 3′-UTR AU content (*Robins and Press, 2005*; *Hausser et al., 2009*), local AU content (*Grimson et al., 2007*; *Nielsen et al., 2007*), minimum distance of the site (*Grimson et al., 2007*), and 3′-UTR length (*Hausser et al., 2009*; *Betel et al., 2010*; *Wen et al., 2011*; *Reczko et al., 2012*). The challenge has been to predict and score site accessibility in a way that is informative after controlling for local AU content, which is important for speaking to the importance of less occlusive secondary structure as opposed to involvement of some AU-binding activity (*Grimson et al., 2007*). The selection of the SA feature in all 1000 bootstrap samples of all four site types showed that it provided discriminatory power apart from that provided by local AU content and other correlated features, which reinforced the idea that the occlusive RNA structure does indeed limit site efficacy. This being said, local AU content, minimum distance of the site, and 3′-UTR length were each also selected in nearly all 1000 bootstrap samples for most site types (*Table 1*), which suggests that either these features were selected for reasons other than their correlation with site accessibility or the definition and scoring of our SA feature has additional room for improvement.

Our ability to confidently identify additional features that each contribute to improved prediction of targeting efficacy was enhanced by our pre-processing of the experimental datasets, which minimized variation from biases unrelated to the sRNA sequence. Yet despite applying this same normalization procedure to our test set, the observed $r^2$ value of 0.14 implied that our model explained only 14% of the variability observed among mRNAs with canonical 7–8 nt 3′-UTR sites (*Figure 4B*). The $r^2$ value increased to 0.15 when considering the usage of alternative 3′-UTR isoforms, but 85% of the variability remained unexplained. Error in the microarray measurements, different sRNA transfection efficiencies, variable incorporation of sRNAs into the silencing complex, and

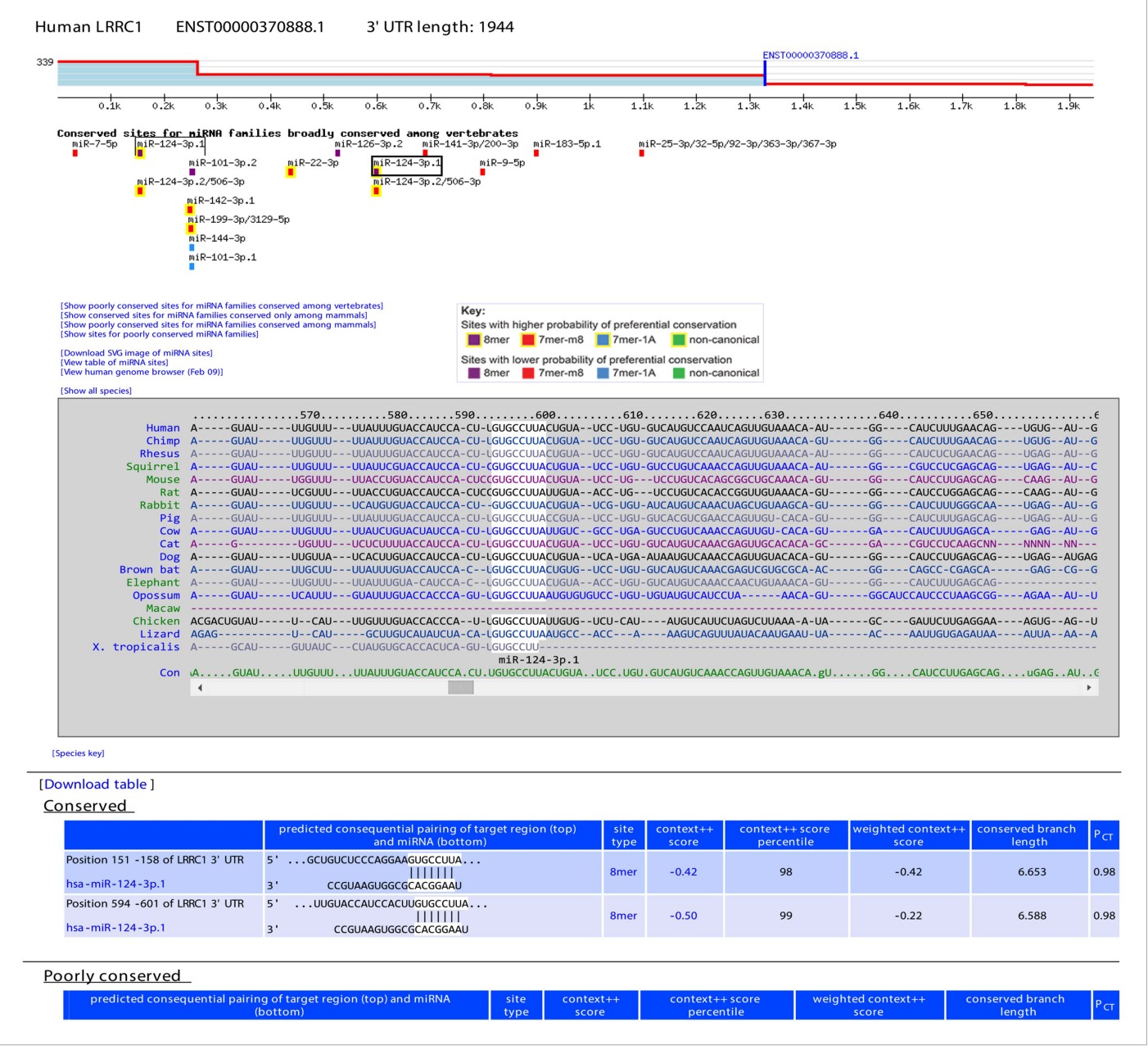

**Figure 7**. Example display of TargetScan7 predictions. The example shows a TargetScanHuman page for the 3′ UTR of the *LRRC1* gene. At the top is the 3′-UTR profile, showing the relative expression of tandem 3′-UTR isoforms, as measured using 3P-seq (***Nam et al., 2014***). Shown on this profile is the end of the longest Gencode annotation (blue vertical line) and the total number of 3P-seq reads (339) used to generate the profile (labeled on the y-axis). Below the profile are predicted conserved sites for miRNAs broadly conserved among vertebrates (colored according to the key), with options to display conserved sites for mammalian conserved miRNAs, or poorly conserved sites for any set of miRNAs. Boxed are the predicted miR-124 sites, with the site selected by the user indicated with a darker box. The multiple sequence alignment shows the species in which an orthologous site can be detected (white highlighting) among representative vertebrate species, with the option to display site conservation among all 84 vertebrate species. Below the alignment is the predicted consequential pairing between the selected miRNA and its sites, showing also for each site its position, site type, context++ score, context++ score percentile, weighted context++ score, branch-length score, and $P_{CT}$ score.

The following figure supplement is available for figure 7:

**Figure supplement 1**. Flowchart of the computational pipeline used to build the TargetScan7 database.

secondary effects of introducing the sRNA presumably made major contributions to the unexplained variability. Nonetheless, imperfections of the context++ model also contributed, raising the question of how much the model might be improved by identifying additional features or developing better methods for scoring and combining existing features. In analyses not described, we evaluated the utility of other types of regression (e.g., linear regression models with interaction terms, lasso/elastic net-regularized regression, multivariate adaptive regression splines, random forest, boosted regression trees, and iterative Bayesian model averaging) and found their performance to be comparable to that of stepwise regression but their resulting models to be considerably more complex and thus less interpretable.

One way to evaluate the extent to which the context++ model might be improved is to consider the degree to which its performance depends on the site-conservation feature. Because sites under selective pressure preferentially possess molecular features required for efficacy, inclusion of the site-conservation feature indirectly recovers some of the information that would otherwise be lost when informative molecular features are missing or imperfectly scored. As more informative molecular features are identified and included in a model, less information remains to be captured, and thus the site-conservation feature cannot contribute as much to the performance of the model. The site-conservation feature ($P_{CT}$) was chosen in all 1000 bootstrap samples of each of the three major site types, which showed that the molecular features of our model still do not fully capture all the determinants under selective pressure. However, $P_{CT}$ was not one of the most informative features (*Figure 4C*). Moreover, when tested as in *Figure 5B*, a model trained on only site type and the other 13 molecular features performed nearly as well as the full context++ model ($r^2$ of 0.126, compared to 0.139 for the full model). This drop in $r^2$ of only 0.013 was substantially less than the 0.044 $r^2$ observed for the site-conservation feature on its own (*Figure 5B*, TargetScan.$P_{CT}$), which suggested that when predicting the response of the test-set mRNAs with the major canonical site types, the context++ model captured 70% (calculated as [0.044–0.013]/0.044) of the information potentially imparted by molecular features.

The relatively minor contribution of site conservation highlights the ability of the context++ model to predict the efficacy of nonconserved sites. Although, everything else being equal, its score for a conserved site is slightly better than that for a nonconserved site, this difference does not prevent inclusion of nonconserved sites from the top predictions. Its general applicability to all canonical sites is useful for evaluating not only nonconserved sites to conserved miRNAs but also all sites for nonconserved miRNAs (e.g., *Figure 6K,L*), including viral miRNAs, as well as the off-targets of synthetic siRNAs and shRNAs.

Our analyses show that recent computational and experimental approaches, including the different types of CLIP, all fail to identify non-canonical targets that are repressed more than control transcripts (*Figures 1, 5C,F*), which reopens the question of whether more than a miniscule fraction of miRNA-mediated repression is mediated through non-canonical sites. Although CLIP approaches can identify non-canonical sites that bind the miRNA with some degree of specificity (*Figure 2*), these non-canonical binding sites do not function to mediate detectable repression. Thus far, the only functional non-canonical sites that can be predicted are 3′-compensatory sites, cleavage sites, and centered sites, which together comprise only a very small fraction (<1%) of the functional sites that can be predicted with comparable accuracy (*Bartel, 2009*; *Shin et al., 2010*). The failure of computational methods to find many functional non-canonical sites cannot rule out the possibility that many of these sites might still exist; if such sites are recognized through unimagined determinants, computational efforts might have missed them. CLIP approaches, on the other hand, provide information that is independent of proposed pairing rules or other hypothesized recognition determinants. Therefore, our analyses of the CLIP results, which detected no residual repression after accounting for canonical interactions, provide the most compelling evidence to date on this issue. Unless there is a substantial technical bias in the CLIP approach (such as a large unanticipated disparity in the propensity of non-canonical interactions to crosslink), the inability of current CLIP approaches to identify non-canonical targets that are repressed more than control transcripts argues strongly against the existence of many functional non-canonical targets.

Why might the CLIP-identified non-canonical sites fail to mediate repression (*Figure 1*) despite binding the miRNA in vivo (*Figure 2*)? Perhaps these sites are ineffective because perfect seed pairing is required for repression. For example, perfect seed pairing might favor binding of a downstream effector, either directly by contributing to its binding site or indirectly through an Argonaute

conformational change that favors its binding. However, this explanation is difficult to reconcile with the activity of 3′-compensatory and centered sites, which can mediate repression despite their lack of perfect seed pairing (*Bartel, 2009*; *Shin et al., 2010*), and the activity of Argonaute artificially tethered to an mRNA, which can mediate repression without any pairing to the miRNA (*Pillai et al., 2004*; *Eulalio et al., 2008*). Therefore, a more plausible explanation is that the CLIP-identified non-canonical sites bind the miRNA too transiently to mediate repression. This explanation for the inefficacy of the recently identified non-canonical sites in the 3′ UTRs resembles that previously proposed for the inefficacy of most canonical sites in ORFs: in both cases the ineffective sites bind to the miRNA very transiently—the canonical sites in ORFs dissociating quickly because of displacement by the ribosome (*Grimson et al., 2007*; *Gu et al., 2009*), and the CLIP-identified non-canonical sites in 3′ UTRs dissociating quickly because they lack both seed pairing and the extensive pairing outside the seed characteristic of effective non-canonical sites (3′-compensatory and centered sites) and thus have intrinsically fast dissociation rates.

The idea that newly identified non-canonical sites bind the miRNA too transiently to mediate repression raises the question of how CLIP could have identified so many of these sites in the first place; shouldn't crosslinking be a function of site occupancy, and shouldn't occupancy be a function of dissociation rates? The answers to these questions partially hinge on the realization that the transcriptome has many more non-canonical binding sites than canonical ones. The motifs identified in the non-canonical interactions have information contents as low as 5.6 bits, and thus are much more common in 3′ UTRs than canonical 6mer or 7mer sites (12 bits and 14 bits, respectively). This high abundance of the non-canonical binding sites would help offset the low occupancy of individual non-canonical sites, such that at any moment more than half of the bound miRNA might reside at non-canonical sites, yielding more non-canonical than canonical sites when using experimental approaches with such high specificity that they can identify a site with only a single read (*Figure 2—figure supplement 1A*).

Although the high abundance of non-canonical sites partly explains why CLIP identifies these sites in such high numbers, it cannot provide the complete answer. Some non-canonical sites in the CLASH and chimera datasets are supported by multiple reads, and all the dCLIP-identified non-canonical sites of the miR-155 study (*Loeb et al., 2012*) are supported by multiple reads. How could some CLIP clusters with ineffective, non-canonical sites have as much read support as some with effective, canonical sites? Our answer to this question rests on the recognition that cluster read density does not perfectly correspond to site occupancy (*Friedersdorf and Keene, 2014*), with the other key factors being mRNA expression levels and crosslinking efficiency. In principle, normalizing the CLIP tag numbers to the mRNA levels minimizes the first factor, preventing a low-occupancy site in a highly expressed mRNA from appearing as well supported as a high-occupancy site in a lowly expressed mRNA (*Chi et al., 2009*; *Jaskiewicz et al., 2012*). Accounting for differential crosslinking efficiencies is a far greater challenge. RNA–protein UV crosslinking is expected to be highly sensitive to the identity, geometry, and environment of the crosslinking constituents, leading to the possibility that the crosslinking efficiency of some sites is orders of magnitude greater than that of others. When considered together with the high abundance of non-canonical sites, variable crosslinking efficiency might explain why so many ineffective non-canonical sites are identified. Overlaying a wide distribution of crosslinking efficiencies onto the many thousands of ineffective, non-canonical sites could yield a substantial number of sites at the high-efficiency tail of the distribution for which the tag support matches that of effective canonical sites. Similar conclusions are drawn for other types of RNA-binding interactions when comparing CLIP results with binding results (*Lambert et al., 2014*).

Variable crosslinking efficiency also explains why many top predictions of the context++ model are missed by the CLIP methods, as indicated by the modest overlap in the CLIP identified targets and the top predictions (*Figure 6*). The crosslinking results are not only variable from site to site, which generates false negatives for perfectly functional sites, but they are also variable between biological replicates (*Loeb et al., 2012*), which imposes a challenge for assigning dCLIP clusters to a miRNA. Although this challenge is mitigated in the CLASH and chimera approaches, which provide unambiguous assignment of the miRNAs to the sites, the ligation step of these approaches occurs at low frequency and presumably introduces additional biases, as suggested by the different profile of non-canonical sites identified by the two approaches (*Figure 2B* and *Figure 2—figure supplement 1A*). For example, CLASH identifies non-canonical pairing to the 3′ region of miR-92 (*Helwak et al., 2013*), whereas the chimera approach identified non-canonical pairing to the 5′ region of this same

miRNA (*Figure 2C*). Because of the false negatives and biases of the CLIP approaches, the context++ model, which has its own flaws, achieves an equal or better performance than the published CLIP studies.

Our observation that CLIP-identified non-canonical sites fail to mediate repression reasserts the primacy of canonical seed pairing for miRNA-mediated gene regulation. Compared to canonical sites, effective non-canonical sites (i.e., 3′-compensatory sites and centered sites) are rare because they require many more base pairs to the miRNA (*Bartel, 2009*; *Shin et al., 2010*) and thus together make up <1% of the effective target sites predicted to date. The requirement of so much additional pairing to make up for a single mismatch to the seed is proposed to arise from several sources. The advantage of propagating continuous pairing past miRNA nucleotide 8 (as occurs for centered sites) might be largely offset by the cost of an unfavorable conformational change (*Bartel, 2009*; *Schirle et al., 2014*). Likewise, the advantage of resuming pairing at the miRNA 3′ region (as occurs for 3′-compensatory sites) might be partially offset by either the relative disorder of these nucleotides (*Bartel, 2009*) or their unfavorable arrangement prior to seed pairing (*Schirle et al., 2014*). In contrast, the seed backbone is pre-organized to favor A-form pairing, with bases of nucleotides 2–5 accessible to nucleate pairing (*Nakanishi et al., 2012*; *Schirle and MacRae, 2012*). Moreover, perfect pairing propagated through miRNA nucleotide 7 creates the opportunity for favorable contacts to the minor groove of the seed:target duplex (*Schirle et al., 2014*).

Our overhaul of the TargetScan website integrated the output of the context++ model with the most current 3′-UTR-isoform data to provide any biologist with an interest in either a miRNA or a potential miRNA target convenient access to the predictions, with an option of downloading code or bulk output suitable for more global analyses. In our continuing efforts to improve the website, several additional functionalities will also soon be provided. To facilitate the exploration of co-targeting networks involving multiple miRNAs (*Tsang et al., 2010*; *Hausser and Zavolan, 2014*), we will provide the option of ranking predictions based on the simultaneous action of several independent miRNA families, to which relative weights (e.g., accounting for relative miRNA expression levels or differential miRNA activity in a cell type of interest) can be optionally assigned. To offer predictions for transcripts not already in the TargetScan database (e.g., novel 3′ UTRs or long non-coding RNAs, including circular RNAs), we will provide a mechanism to compute context++ scores interactively for a user-specified transcript. Likewise, to offer predictions for a novel sRNA sequence (e.g., off-target predictions for an siRNA), we will provide a mechanism to retrieve context++ scores interactively for a user-specified sRNA. To visualize the expression signature that results from perturbing a miRNA, we will provide a tool for the user to input mRNA/protein fold changes from high-throughput experiments and obtain a cumulative distribution plot showing the response of predicted targets relative to that of mRNAs without sites. Thus, with the current and future improvements to TargetScan, we hope to enhance the productivity of miRNA research and the understanding of this intriguing class of regulatory RNAs.

## Materials and methods

### Microarray, RNA-seq, and RPF dataset processing

A list of microarray, RNA-seq, ribosome profiling, and proteomic datasets used for analyses, as well as the corresponding figures in which they were used, is provided (*Table 2*). We considered developing the model using RNA-seq data rather than microarray data, but microarray datasets were still much more plentiful and were equally suitable for measuring the effects of sRNAs. Unless pre-processed microarray data were provided by previous studies (as indicated in *Table 2*), raw data were processed using Bioconductor release 2.14 in the R programming language v3.1.1 (*Gentleman et al., 2004*; *Development Core Team, 2014*). Affymetrix data were first background-corrected with the 'gcrma' R package (*Wu et al., 2004*), whereas Illumina BeadArray data from the miR-302 knockdown and miR-522 transfection datasets (*Lipchina et al., 2011*; *Tan et al., 2014*) were processed and background-corrected using the 'lumiR' and 'lumiExpresso' functions in the 'lumi' R package (*Du et al., 2008*). A robust linear regression model was then used to fit to the probe intensities using the 'lmFit' function (parameter 'method = 'robust'') in the 'limma' R package v3.6.9 (*Smyth, 2004*, *2005*), computing differential expression information with the provided eBayes function. Probe IDs were then converted to RefSeq or Ensembl IDs (e.g., using the hgu133plus2ENSEMBL and IlluminaID2nuID/lumiHumanAllENSEMBL functions to convert Affymetrix and BeadArray probe IDs, respectively), and the fold change for each

**Table 2.** Summary of datasets analyzed in this study, and corresponding figures using the datasets

| Figure | Gene expression omnibus (GEO) ID, ArrayExpress ID, or data source | Reference |
|---|---|---|
| *Figure 1A*, *Figure 1—figure supplement 4A* | GSM854425, GSM854430, GSM854431, GSM854436, GSM854437, GSM854442, GSM854443 | (*Bazzini et al., 2012*) |
| *Figure 1B*, *Figure 6B* | GSM1012118, GSM1012119, GSM1012120, GSM1012121, GSM1012122, GSM1012123 | (*Loeb et al., 2012*) |
| *Figure 1C*, *Figure 1—figure supplement 2A*, *Figure 6C,D* | E-TABM-232 | (*Rodriguez et al., 2007*) |
| *Figure 1D,F* | GSM1122217, GSM1122218, GSM1122219, GSM1122220, GSM1122221, GSM1122222, GSM1122223, GSM1122224, GSM1122225, GSM1122226 | (*Helwak et al., 2013*) |
| *Figure 1E*, *Figure 1—figure supplement 3A–D*, *Figure 6I,J* | GSM538818, GSM538819, GSM538820, GSM538821 | (*Hafner et al., 2010*) |
| *Figure 1G* | GSM156524, GSM156532, GSM210897, GSM210898, GSM210901, GSM210903, GSM210904, GSM210907, GSM210909, GSM210911, GSM210913, GSM37599, http://psilac.mdc-berlin.de/download/ (let7b_32h, miR-30_32h, miR-155_32h, miR-16_32h) | (*Lim et al., 2005*; *Grimson et al., 2007*; *Linsley et al., 2007*; *Selbach et al., 2008*) |
| *Figure 1H*, *Figure 6K,L* | E-MTAB-2110 | (*Tan et al., 2014*) |
| *Figure 1I*, *Figure 1—figure supplement 2B*, *Figure 6E* | GSM1479572, GSM1479576, GSM1479580, GSM1479584 | (*Eichhorn et al., 2014*) |
| *Figure 1—figure supplement 1A* | GSM210897, GSM210898, GSM210901, GSM210903, GSM210904, GSM210907, GSM210909, GSM210911, GSM210913, GSM37599, GSM37601 | (*Lim et al., 2005*; *Grimson et al., 2007*) |
| *Figure 1—figure supplement 1B*, *Figure 3*, *Figure 3—figure supplement 1B,C*, *Figure 4* | 74 datasets compiled in Supplementary data 4 of *Garcia et al. (2011)*, used as is or after normalization (*Supplementary file 1*); GSM119707, GSM119708, GSM119710, GSM119743, GSM119745, GSM119746, GSM119747, GSM119749, GSM119750, GSM119759, GSM119761, GSM119762, GSM119763, GSM133685, GSM133689, GSM133699, GSM133700, GSM134325, GSM134327, GSM134466, GSM134480, GSM134483, GSM134485, GSM134511, GSM134512, GSM134551, GSM210897, GSM210898, GSM210901, GSM210903, GSM210904, GSM210907, GSM210909, GSM210911, GSM210913, GSM37599, GSM37601; E-MEXP-1402 (1595297366, 1595297383, 1595297389, 1595297394, 1595297399, 1595297422, 1595297427, 1595297432, 1595297491, 1595297496, 1595297501, 1595297507, 1595297513, 1595297518, 1595297524, 1595297530, 1595297535, 1595297564, 1595297588, 1595297595, 1595297605, 1595297614, 1595297621, 1595297627, 1595297644, 1595297650, 1595297662); E-MEXP-668 (16012097016666, 16012097016667, 16012097016668, 16012097016669, 16012097017938, 16012097017939, 16012097017952, 16012097017953, 16012097018568, 251209725411) | (*Lim et al., 2005*; *Birmingham et al., 2006*; *Schwarz et al., 2006*; *Jackson et al., 2006a*; *Jackson et al., 2006b*; *Grimson et al., 2007*; *Anderson et al., 2008*) |
| *Figure 1—figure supplement 1C* | GSM95614, GSM95615, GSM95616, GSM95617, GSM95618, GSM95619 | (*Giraldez et al., 2006*) |
| *Figure 1—figure supplement 1D,F* | GSM1269344, GSM1269345, GSM1269348, GSM1269349, GSM1269350, GSM1269351, GSM1269354, GSM1269355, GSM1269356, GSM1269357, GSM1269360, GSM1269361, GSM1269362, GSM1269363 | (*Nam et al., 2014*) |
| *Figure 1—figure supplement 3E*, *Figure 6H* | http://icb.med.cornell.edu/faculty/betel/lab/betelab_v1/Data.html | (*Lipchina et al., 2011*) |
| *Figure 1—figure supplement 4B* | http://psilac.mdc-berlin.de/media/database/release-1.0/protein/pSILAC_all_protein_ratios_OE.txt (miR155) | (*Selbach et al., 2008*) |
| *Figure 3—figure supplement 1A* | GSM416753 | (*Mayr and Bartel, 2009*) |
| *Figure 5*, *Figure 5—figure supplement 1* | GSM156522, GSM156580, GSM156557, GSM156548, GSM156533, GSM156532, GSM156524, processed and normalized (*Supplementary file 2*) | (*Linsley et al., 2007*) |
| *Figure 6A* | GSM37601 | (*Lim et al., 2005*) |
| *Figure 6F,G* | GSM363763, GSM363766, GSM363769, GSM363772, GSM363775, GSM363778 | (*Hausser et al., 2009*) |

mRNA was computed as the median fold change for all probes corresponding to the mRNA. Finally, because about half of the genes in the genome were either not expressed in the cell type examined, or were expressed at a level that was so close to the background that they were prone to have noisy fold-change measurements, the following filters were applied:

i. For microarray datasets examining the effect of either knocking down either miR-92 or 25 miRNA families in HEK293 cells (*Hafner et al., 2010*; *Helwak et al., 2013*), transfecting miR-7 or miR-124 into HEK293 cells (*Hausser et al., 2009*), knocking out miR-155 in T$_h$1 or T$_h$2 cells (*Rodriguez et al., 2007*), or transfecting each of the 7 miRNAs in HCT116 cells (*Linsley et al., 2007*), we computed the mean signal for each mRNA (averaging the signal with and without the miRNA), and retained mRNAs exceeding the median of this distribution.

ii. For microarray datasets examining the effect of injecting miR-430 into MZDicer embryos (*Giraldez et al., 2006*) or knocking out miR-155 in T cells (*Loeb et al., 2012*), we required the mean signal intensity of an mRNA to exceed 3.0 and 2.5, respectively.

iii. For Illumina BeadArray datasets examining the effect of either knocking down miR-302/367 (*Lipchina et al., 2011*) or transfecting miR-522 (*Tan et al., 2014*), we required the mean signal intensity to exceed 7.5 and 7.0, respectively.

iv. For all 74 small-RNA transfections of HeLa cells, we required mRNA expression levels to exceed 10 reads per million (RPM), as quantified by RNA-seq in mock-transfected HeLa cells (*Guo et al., 2010*).

v. For analysis of RNA-seq or RPF datasets examining the effect of either losing Dicer in zebrafish embryos (*Bazzini et al., 2012*), transfecting miR-124 into HEK293, HeLa, or Huh7 cells (*Nam et al., 2014*), or knocking out miR-155 in B cells (*Eichhorn et al., 2014*), we required mRNA expression levels to exceed 10 RPM, as quantified in the condition lacking the perturbed miRNA.

vi. For analysis of proteomic results, we used the pre-computed data provided in the table of significantly detectable peptides (*Selbach et al., 2008*).

These thresholds were chosen based upon visual inspection of plots evaluating the relationship between mean expression level and fold change (commonly known as 'MA plots' in the context of microarrays), attempting to balance the tradeoff between maximal sample size and reduced noise. The overall conclusions were robust to the choice of the threshold. After imposing the threshold, all fold-change values were centered by subtracting the median fold-change value of the 'no-site' mRNAs in each sRNA perturbation experiment, except in the case of *Figure 5—figure supplement 1B,C*, in which data were mean-centered.

## Crosslinking and other interactome datasets

When available, target genes identified using high-throughput CLIP data were collected from the supplemental materials of the corresponding studies (*Lipchina et al., 2011*; *Loeb et al., 2012*; *Helwak et al., 2013*; *Grosswendt et al., 2014*). For the original PAR-CLIP study (*Hafner et al., 2010*), targets were inferred from an online resource of all endogenous HEK293 clusters (http://www.mirz.unibas.ch/restricted/clipdata/RESULTS/CLIP_microArray/Antago_mir_vs_ALL_AGO.txt) as well as clusters observed after transfection of either miR-7 (http://www.mirz.unibas.ch/restricted/clipdata/RESULTS/miR7_TRANSFECTION/miR7_TRANSFECTION.html) or miR-124 (http://www.mirz.unibas.ch/restricted/clipdata/RESULTS/miR124_TRANSFECTION/miR124_TRANSFECTION.html). For dCLIP-supported miR-124 sites identified in the original high-throughput CLIP study (*Chi et al., 2009*), we used clusters whose genomic coordinates were provided by SW Chi (*Supplementary file 3*), extracting the corresponding sequences using the 'getfasta' utility in BEDTools v2.20.1 (parameters '-s -name -tab ') (*Quinlan and Hall, 2010*). When evaluating the function of non-canonical sites supported by CLIP or IMPACT-seq (*Figure 1* and *Figure 1—figure supplements 1–4*), a cluster (or CLASH/chimera interaction) with a 6–8mer site (but not only an offset-6mer site, unless otherwise indicated in the figure legends) corresponding to the cognate miRNA was classified as harboring a canonical site. Otherwise, the cluster (or CLASH/chimera interaction) was classified as containing a non-canonical site, and the corresponding mRNA was carried forward for functional evaluation as a non-canonical CLIP-supported target if it also had no cognate 6–8mer sites (but allowing offset-6mer sites) in its 3′ UTR (using either RefSeq or Ensembl 3′-UTR annotations as appropriate for the gene IDs published by the CLIP study). When comparing the response of canonical CLIP-supported targets to that of TargetScan7 predictions (*Figure 6*), the canonical CLIP-supported sites were additionally required to

fall within (and on the same DNA strand as) annotated 3′ UTRs, as evaluated by the intersectBED utility in BEDTools v2.20.1 (parameter '-s') (*Quinlan and Hall, 2010*).

## Motif discovery for non-canonical binding sites

To identify non-canonical modes of binding, all CLASH interactions assigned to a particular miRNA family (defined as all mature miRNA sequences sharing a common sequence in nucleotide positions 2–8) were collected. Interactions containing the cognate canonical site type (offset 6mer, 6mer, 7mer-m8, 7mer-A1, or 8mer) were removed. For all miRNA families with at least 50 unique CLASH interactions remaining, enriched motifs were evaluated using MEME version 4.9.0 (parameters '-p 100 -dna -mod zoops -nmotifs 10 -minw 4 -maxw 8 -maxsize 1,000,000,000') (*Bailey and Elkan, 1994*). All motifs with an E-value $< 10^{-3}$ are reported along with their E-values rounded to the nearest log-unit. Instances in which a top-ranked motif exceeded this E-value were also reported if the motif was an approximate complementary match to the miRNA. For each miRNA family, the top motif identified by MEME was aligned to a representative mature miRNA using FIMO (parameters '–norc–motif 1 –thresh 0.01') (*Grant et al., 2011*), considering the reverse complement of the mature miRNA with the last nucleotide of this reverse complement changed to an A (to capture the enrichment of an adenosine across from the 5′ nucleotide of a miRNA, as occurs in 8mer and 7mer-A1 sites). Logos were also manually examined to determine if any mapped to the mature miRNA with a bulged nucleotide. The same procedure was performed for chimera interactions, for dCLIP clusters reported for miR-124 and miR-155, and for IMPACT-seq clusters reported for miR-522.

## Microarray dataset normalization

For each of the 74 transfection experiments of the compendium (*Table 2*), data were first partitioned into the mRNA fold changes ($\log_2$) measured in the given experiment (the response variable) as well as a matrix of the corresponding mRNA fold changes for the remaining 73 datasets (the predictor variables). A PLSR model was then trained to predict the response using information from the predictor variables. When training the model, PLSR took into account the correlated structure of the predictor matrix, decomposing it into a low-dimensional representation that maximally explained the response variable.

Stating the procedure more formally, let Z be an $n \times m$ matrix consisting of $\log_2$(mRNA fold change) measurements of $n$ mRNAs in response to the sRNA transfected in each of $m$ experiments. Let $y_i$ represent measurements for all mRNAs in the $i$th experiment of Z, and $X_{\bar{i}}$ represent measurements for all mRNAs from all experiments except for the $i$th experiment in Z. Finally, let $T_{\bar{i}}$ be a matrix with identical dimensions as $X_{\bar{i}}$, with entries $t_{j,k} = 1$ if the 3′ UTR of mRNA $j$ in $X_{\bar{i}}$ contains a canonical 7–8 nt match to the small RNA transfected in experiment $k$ in $X_{\bar{i}}$, and $t_{j,k} = 0$ otherwise. Missing values in Z represent cases in which the mRNA signal in the microarray was too low to be reliably measured. The following algorithm was used to normalize each $y_i$ for $i \in \{1…74\}$:

i. For values in $T_{\bar{i}}$ in which $t_{j,k} = 1$, the corresponding value $x_{j,k}$ in $X_{\bar{i}}$ was removed, which prevented the loss of signal in $y_{i,j}$ due to sRNA-mediated regulation of the mRNA in two independent experiments.

ii. mRNAs in $y_i$, $X_{\bar{i}}$, and $T_{\bar{i}}$ were removed if the $\log_2$(mRNA fold change) was either undefined in $y_i$ or undefined in greater than 50% of experiments in $X_{\bar{i}}$.

iii. For the remaining missing values in $X_{\bar{i}}$, values were imputed using the $k$-nearest neighbors algorithm, using $k = 20$, as implemented in the *impute.knn* function in the 'impute' R package (*Troyanskaya et al., 2001*). Results were robust to the choice of imputation algorithm (data not shown).

iv. To remove biases afflicting $y_i$, $y_i$ was predicted from $X_{\bar{i}}$ using partial least squares regression, as implemented in the *plsr* function in the 'pls' R package (*Mevik and Wehrens, 2007*). Ten-fold cross-validation was used to choose an appropriate number of components in the regression. Values of $y_i$ were then adjusted to their residuals as such: $y_i \leftarrow y_i - \hat{y}_i$, where $\hat{y}_i$ was the vector of predicted values of $y_i$ from the regression (*Supplementary file 1*).

An analogous normalization procedure was performed for each of the seven transfection experiments of the test set (*Supplementary file 2*).

## RNA structure prediction

3′ UTRs were folded locally using RNAplfold (*Bernhart et al., 2006*), allowing the maximal span of a base pair to be 40 nucleotides, and averaging pair probabilities over an 80 nt window (parameters -L

40 -W 80), parameters found to be optimal when evaluating siRNA efficacy (*Tafer et al., 2008*). For each position 15 nt upstream and downstream of a target site, and for 1–15 nt windows beginning at each position, the partial correlation of the $\log_{10}$(unpaired probability) to the $\log_2$(mRNA fold change) associated with the site was plotted, controlling for known determinants of targeting used in the context+ model, which include min_dist, local_AU, 3P_score, SPS, and TA (*Garcia et al., 2011*). For the final predicted SA score used as a feature, we computed the $\log_{10}$ of the probability that a 14-nt segment centered on the match to sRNA positions 7 and 8 was unpaired.

## Calculation of $P_{CT}$ scores

We updated human $P_{CT}$ scores using the following datasets: (i) 3′ UTRs derived from 19,800 human protein-coding genes annotated in Gencode version 19 (*Harrow et al., 2012*), and (ii) 3′-UTR multiple sequence alignments (MSAs) across 84 vertebrate species derived from the 100-way multiz alignments in the UCSC genome browser, which used the human genome release hg19 as a reference species (*Kent et al., 2002*; *Karolchik et al., 2014*). We used only 84 of the 100 species because, with the exception of coelacanth (a lobe-finned fish more related to the tetrapods), the fish species were excluded due to their poor quality of alignment within 3′ UTRs. Likewise, we updated the mouse scores using: (i) 3′ UTRs derived from 19,699 mouse protein-coding genes annotated in Ensembl 77 (*Flicek et al., 2014*), and (ii) 3′-UTR MSAs across 52 vertebrate species derived from the 60-way multiz alignments in the UCSC genome browser, which used the mouse genome release mm10 as a reference species (*Kent et al., 2002*; *Karolchik et al., 2014*). As before, we partitioned 3′ UTRs into ten conservation bins based upon the median branch-length score (BLS) of the reference-species nucleotides (*Friedman et al., 2009*). However, to estimate branch lengths of the phylogenetic trees for each bin, we concatenated alignments within each bin using the 'msa_view' utility in the PHAST package v1.1 (parameters '–unordered-ss–in-format SS–out-format SS–aggregate $species_list–seqs $species_subset', where $species_list contains the entire species tree topology and $species_subset contains the topology of the subtree spanning the placental mammals) (*Siepel and Haussler, 2004*). We then fit trees for each bin using the 'phyloFit' utility in the PHAST package v1.1, utilizing the generalized time-reversible substitution model and a fixed-tree topology provided by UCSC (parameters '-i SS–subst-mod REV–tree $tree', where $tree is the Newick format tree of the placental mammals) (*Siepel and Haussler, 2004*). $P_{CT}$ parameters and scores were then calculated as described, estimating the signal of conservation for each seed family relative to that of its corresponding 50 control $k$-mers, matched for $k$-mer length and rate of dinucleotide conservation at varying branch-length windows (*Friedman et al., 2009*). All phylogenetic trees and $P_{CT}$ parameters are available for download at the TargetScan website (targetscan.org).

## Selection of mRNAs for regression modeling

The mRNAs were selected to avoid those from genes with multiple highly expressed alternative 3′-UTR isoforms, which would have otherwise obscured the accurate measurement of features such as len_3UTR or min_dist, and also created situations in which the response was diminished because some isoforms lacked the target site. HeLa 3P-seq results (*Nam et al., 2014*) were used to identify genes in which a dominant 3′-UTR isoform comprised ≥90% of the transcripts (*Supplementary file 1*). For each of these genes, the mRNA with the dominant 3′-UTR isoform was carried forward, together with the ORF and 5′-UTR annotations previously chosen from RefSeq (*Garcia et al., 2011*). Sequences of these mRNA models are provided as Supplemental material at http://bartellab.wi.mit.edu/publication.html. To prevent the presence of multiple 3′-UTR sites to the transfected sRNA from confounding attribution of an mRNA change to an individual site, these mRNAs were further filtered within each dataset to consider only mRNAs that contained a single 3′-UTR site (either an 8mer, 7mer-m8, 7mer-A1, or 6mer) to the cognate sRNA.

## Scaling the scores of each feature

Features that exhibited skewed distributions, such as len_5UTR, len_ORF, and len_3UTR were $\log_{10}$ transformed (*Table 1*), which made their distributions approximately normal. These and other continuous features were then normalized to the (0, 1) interval as described (e.g., see Supplementary Figure 5 in *Garcia et al., 2011*), except a trimmed normalization was implemented to prevent outlier values from distorting the normalized distributions. For each value, the 5th percentile of the feature was subtracted

from the value, and the resulting quantity was divided by the difference between the 95th and 5th percentiles of the feature. Percentile values are provided for the subset of continuous features that were scaled (*Table 3*). The trimmed normalization facilitated comparison of the contributions of different features to the model, with absolute values of the coefficients serving as a rough indication of their relative importance.

## Stepwise regression and multiple linear regression models

We generated 1000 bootstrap samples, each including 70% of the data from each transfection experiment of the compendium of 74 datasets (*Supplementary file 1*), with the remaining data reserved as a held-out test set. For each bootstrap sample, stepwise regression, as implemented in the *stepAIC* function from the 'MASS' R package (*Venables and Ripley, 2002*), was used to both select the most informative combination of features and train a model. Feature selection maximized the Akaike information criterion (AIC), defined as: $-2 \ln(L) + 2k$, where $L$ was the likelihood of the data given the linear regression model and $k$ was the number of features or parameters selected. The 1000 resulting models were each evaluated based on their $r^2$ to the corresponding test set. To illustrate the utility of adding features not included in our previous models, these $r^2$ values were compared to those obtained when re-training the multiple linear regression coefficients on each bootstrap sample using only the features of either the context-only or the context+ model, and computing $r^2$ values on the corresponding test sets. The stepwise regression was implemented independently for each of the site types, and a final set of features was chosen as those that were selected for at least 99% of the bootstrap samples of at least two site types. Using this group of features and the entire compendium of 74 datasets as a training set, we trained a multiple linear regression model for each site type (*Figure 4—source data 1*). As done previously for TargetScan6 predictions, scores for 8mer, 7mer-m8, 7mer-A1, and 6mer sites were bounded to be no greater than −0.03, −0.02, −0.01, and 0, respectively, thereby creating a piece-wise linear function for each site type.

## Collection and processing of previous predictions

To compare predictions from different miRNA target prediction tools, we collected the following freely downloadable predictions: AnTar (predictions from either miRNA-transfection or CLIP-seq models) (*Wen et al., 2011*), DIANA-microT-CDS (September 2013) (*Reczko et al., 2012*), ElMMo v5 (January 2011) (*Gaidatzis et al., 2007*), MBSTAR (all predictions) (*Bandyopadhyay et al., 2015*), miRanda-MicroCosm v5 (*Griffiths-Jones et al., 2008*), miRmap v1.1 (September 2013) (*Vejnar and Zdobnov, 2012*), mirSVR (August 2010) (*Betel et al., 2010*), miRTarget2 (from miRDB v4.0, January 2012) (*Wang, 2008*; *Wang and El Naqa, 2008*), MIRZA-G (sets predicted either with or without conservation features and either with or without more stringent seed-match requirements, March

**Table 3**. Scaling parameters used to normalize data to the (0, 1) interval

| Feature | 8mer | | 7mer-m8 | | 7mer-A1 | | 6mer | |
|---|---|---|---|---|---|---|---|---|
| | 5th % | 95th % | 5th % | 95th % | 5th % | 95th % | 5th % | 95th % |
| 3P_score | 1.000 | 3.500 | 1.000 | 3.500 | 1.000 | 3.500 | 1.000 | 3.500 |
| SPS | −11.130 | −5.520 | −11.130 | −5.490 | −8.410 | −3.330 | −8.570 | −3.330 |
| TA_3UTR | 3.113 | 3.865 | 3.067 | 3.887 | 3.145 | 3.887 | 3.113 | 3.887 |
| Len_3UTR | 2.392 | 3.637 | 2.409 | 3.615 | 2.413 | 3.630 | 2.405 | 3.620 |
| Len_ORF | 2.788 | 3.753 | 2.773 | 3.729 | 2.773 | 3.730 | 2.775 | 3.731 |
| Min_dist | 1.415 | 3.113 | 1.491 | 3.096 | 1.431 | 3.117 | 1.477 | 3.106 |
| Local_AU | 0.308 | 0.814 | 0.277 | 0.782 | 0.342 | 0.801 | 0.295 | 0.772 |
| SA | −4.356 | −0.661 | −5.218 | −0.725 | −4.230 | −0.588 | −5.082 | −0.666 |
| $P_{CT}$ | 0.000 | 0.816 | 0.000 | 0.364 | 0.000 | 0.449 | 0.000 | 0.193 |

Provided are the 5th and 95th percentile values for continuous features that were scaled, after the values of the feature were appropriately transformed as indicated (*Table 1*).

2015) (*Gumienny and Zavolan, 2015*), PACCMIT-CDS (sets predicted either with or without conservation features) (*Marin et al., 2013*), PicTar2 (from the doRiNA web resource; sets conserved to either fish, chicken, or mammals) (*Krek et al., 2005*; *Anders et al., 2012*), PITA Catalog v6 (3/15 flank for either 'All' or 'Top' predictions, August 2008) (*Kertesz et al., 2007*), RNA22 (May 2011) (*Miranda et al., 2006*), SVMicrO (February 2011) (*Liu et al., 2010*), TargetRank (all scores from web server) (*Nielsen et al., 2007*), TargetSpy (all predictions) (*Sturm et al., 2010*), TargetScan v5.2 (either conserved or all predictions, June 2011) (*Grimson et al., 2007*), and TargetScan v6.2 (either conserved predictions ranked by the context+ model or all predictions ranked by either the context+ model or $P_{CT}$ scores, June 2012) (*Friedman et al., 2009*; *Garcia et al., 2011*). For algorithms providing site-level predictions (i.e., ElMMo, MBSTAR, miRSVR, PITA, and RNA22), scores were summed within genes or transcripts (if available) to acquire an aggregate score. For algorithms providing multiple transcript-level predictions (i.e., miRanda-MicroCosm, PACCMIT-CDS, and TargetSpy), the transcript with the best score was selected as the representative transcript isoform. In all cases, predictions with gene symbol or Ensembl ID formats were translated into RefSeq format. When computing $r^2$ to the test sets, mRNAs that were not predicted by the algorithm to be a target were assigned the worst score in the range of all scores generated by the algorithm.

## 3′-UTR profiles for TargetScan7 predictions

To build databases of human and mouse 3′-UTR profiles, we began with the 'basic' set of protein-coding gene models deposited in Gencode v19 (human hg19 assembly) and Gencode vM3 (mouse mm10 assembly), respectively (*Harrow et al., 2012*). For each unique stop codon in each set of gene models, we selected the transcript with the longest 3′ UTR as its representative transcript. If other datasets indicated that the 3′ UTRs of these representative transcripts have longer tandem isoforms, we extended them accordingly, using additional annotations provided by (i) the 'comprehensive' set of Gencode gene models (*Harrow et al., 2012*), (ii) all mRNAs in the RefSeq database (*Pruitt et al., 2012*), downloaded from the refGene database through the UCSC table browser (*Kent et al., 2002*), and (iii) 3′-UTR extensions supported by RNA-seq evidence (*Miura et al., 2013*), after transforming mm9 to mm10 coordinates using liftOver (*Hinrichs et al., 2006*). We then used 3P-seq clusters from human and mouse (*Nam et al., 2014*) (again after transforming coordinates with liftOver) to further extend 3′ UTRs when possible, searching within a 5400 nt region downstream of the stop codon (excluding the regions containing annotated introns) for a cleavage and polyadenylation site supported by at least one 3P-seq cluster, prohibiting the search to extend beyond the start position of any annotated downstream exon. The 5400 nt window was chosen because the 99th percentile of the lengths of previously annotated mouse and human 3′ UTRs was ~5400 nt. Zebrafish 3′ UTRs for TargetScanFish were identical to those annotated previously (*Ulitsky et al., 2012*). For each representative transcript, 3P-seq clusters mapping within the extended 3′ UTR were used to quantify the relative levels of alternative tandem isoforms, thereby generating a 3′-UTR profile. For human and mouse transcripts, all 3P-seq datasets for cell lines/tissues of each species were combined, after normalizing for the sequencing depth (i.e., number of uniquely mapping tags) of each dataset, to generate meta profiles. To perform this normalization, the number of tags overlapping the 3′ UTR of each annotated transcript was first summed. A matrix of summed tag counts for each cell line/tissue and for each transcript was then compiled, removing transcripts with no tags in any cell type. This matrix was then upper-quartile normalized by calculating the 75th quantile of counts in each cell type, using the calcNormFactors function (parameter 'method = 'upperquartile'') in the 'edgeR' R package (*Robinson et al., 2010*). These scaling factors were then applied to all tags, and the normalized tag counts corresponding to each 3P-seq cluster from different cell lines/tissues were summed. A pseudocount of 0.1 tag was assigned to the longest tandem 3′-UTR isoform, which accommodated cases in which the longest annotated 3′ UTR did not have tag support. In addition, 5 pseudocounts were assigned to the longest Gencode 3′-UTR isoform, which gave preference to this Gencode annotation if the UTR had poor 3P-seq coverage. The 3′-UTR profiles were then generated and used to compute the affected isoform ratio (AIR) and weighted context++ score for each predicted target site as depicted in *Figures 2A, 3A*, respectively, of *Nam et al. (2014)*. For zebrafish transcripts, profiles were generated for each developmental stage with a 3P-seq dataset. All input and output annotation files as well as scripts are available for download at TargetScan (targetscan.org).

## MicroRNA sets for TargetScan7

When partitioning miRNA families according to their conservation level, we began with a high-confidence set of human miRNAs supported by small-RNA sequencing (T Tuschl, personal communication) that shared nucleotides 2–8 with a mouse miRNA supported by small-RNA sequencing (*Chiang et al., 2010*). We then extracted 100-way multiz alignments of each mature miRNA from the UCSC Genome Browser and counted the number of species for which nucleotides 2–8 of the miRNA did not change. As an initial pass, those conserved among ≥40 species were classified as mammalian conserved, and those conserved among >60 species were classified as more broadly conserved among vertebrate species. Due to poorer quality alignments for more distantly related species, this procedure misclassified several more broadly conserved miRNAs as mammalian conserved. Therefore, mammalian conserved miRNAs that aligned with >90% homology to a mature miRNA from chicken, frog, or zebrafish, as annotated in miRBase release 21 (*Kozomara and Griffiths-Jones, 2014*), were re-classified as more broadly conserved. In addition, miR-489 was included in the broadly conserved set of TargetScanHuman (but not TargetScanMouse) despite having a seed substitution in mouse.

Some mammalian pri-miRNAs give rise to two or three abundant miRNA isoforms that have different seeds, either because both strands of the miRNA duplex load into Argonaute with near-equal efficiencies or because processing heterogeneity gives rise to alternative 5′ termini (*Azuma-Mukai et al., 2008*; *Morin et al., 2008*; *Wu et al., 2009*; *Chiang et al., 2010*). To annotate these abundant alternative isoforms, we identified all isoforms expressed at ≥33% of the level of the most abundant isoform, as determined from high-throughput sequencing (allowing for 3′ heterogeneity within each isoform). These isoforms were carried forward as mammalian conserved isoforms if they also satisfied this property in the mouse small-RNA sequencing data (*Chiang et al., 2010*), and as broadly conserved isoforms if they satisfied this property in zebrafish small-RNA sequencing data available in miRBase release 21. Adhering to the miRNA naming convention, if two isoforms mapped to the 5′ and 3′ arms of the hairpin they were named '–5p' and '–3p', respectively, and if two isoforms were processed from the same arm they were named '.1' and '.2' in decreasing order of their abundance, as detected in the human.

All mature miRNAs were downloaded from miRBase release 21 (*Kozomara and Griffiths-Jones, 2014*). Those that matched a conserved miRNA at nucleotides 2–8 were considered part of that miRNA family. All miRNAs and miRNA isoforms annotated in miRBase but not meeting our criteria for conservation in mammals or beyond were also grouped into families based on the identity of nucleotides 2–8 and were classified as poorly conserved miRNAs (which included many small RNAs misclassified as miRNAs). The miRNA seed families and associated conservation classifications are available for download at TargetScan (targetscan.org).

## TargetScan7 predictions

TargetScan (v7.0) provides the option of ranking predicted targets of mammalian miRNAs according to either cumulative weighted context++ score (CWCS), which ranks based upon the predicted repression, or aggregate $P_{CT}$ score of the longest 3′-UTR isoform, which ranks based upon the confidence that targeting is evolutionarily conserved (*Figure 7—figure supplement 1*).

For each predicted target, the CWCS estimated the total repression expected from multiple sites to the same miRNA. This score was calculated using the 3′-UTR profiles to weight the marginal effect of each additional site to the miRNA while also taking into account the predicted mRNA depletion resulting from any downstream sites to the same miRNA. This approach was improved over that we used previously to calculate total wContext+ scores (*Nam et al., 2014*), in that it did not over-estimate the aggregate effect of multiple sites in distal isoforms. For each miRNA family, 8mer, 7mer-m8, 7mer-A1, and 6mer sites were first filtered to remove overlapping sites, and for each reference 3′ UTR, nonoverlapping sites to the same miRNA were numbered from 1 to $n$, starting at the distal end of the 3′ UTR. For each site $i$, from 1 to $n$, the cumulative predicted repression at that site ($C_i$) was calculated as $C_i = C_{(i-1)} + (1 - 2^{CS_i})(\text{AIR}_i - C_{(i-1)})$, in which $CS_i$ and $\text{AIR}_i$ were the context++ score and AIR of site $i$, and the $(1 - 2^{CS_i})(\text{AIR}_i - C_{(i-1)})$ term predicted the marginal repression of site $i$, in which the predicted repression at the site $(1 - 2^{CS_i})$ was modified based on the fraction of mRNAs containing that site ($\text{AIR}_i$) as reduced by the mRNA depletion predicted to occur from the action of any more distal sites ($C_{(i-1)}$, assigning $C_0$ as 0). The CWCS was then calculated as $\log_2(1 - C_n)$, in which $C_n$ was

the $C_i$ at the most proximal site of the reference 3′ UTR. For each reference 3′ UTR, CWCSs were calculated for each member of a miRNA family, and the score from the member with the greatest predicted repression was chosen to represent that family, and the reference 3′ UTR with the most 3P-seq tags was chosen to represent the gene.

When scoring features that can vary with 3′-UTR length (Min_dist, Len_3UTR, and Off6m), a weighted score was used that accounted for the abundance of each 3′-UTR tandem isoform in which the site existed, as estimated from a compendium of 3P-seq datasets from the same species (*Nam et al., 2014*). Although 6mer sites are used to calculate cumulative weighted context++ scores, and 6mer sites are tallied in the tables, the locations of these 6mer sites are not displayed, and targets with only 6mer sites are not listed. When calculating $P_{CT}$ scores, the most abundant 3′-UTR isoform as defined by 3P-seq was used to determine the conservation bin to which the 3′ UTR belonged. Sites corresponding to poorly conserved and mammalian-conserved miRNA seed families or sites overlapping annotated ORF regions were assigned $P_{CT}$ scores of zero. For TargetScanFish, genome-wide alignment quality in zebrafish 3′ UTRs was not of sufficient quality to compute $P_{CT}$ scores, so a $P_{CT}$ value of zero was assigned to all sites when computing context++ scores. All $P_{CT}$ parameters and parameters for tree branch lengths and regression models, along with pre-computed context++ scores for human, mouse, zebrafish, and other vertebrate species are available for download (targetscan.org). Perl scripts using these parameters to compute context++ scores, weighted context++ scores, CWCSs, and aggregate $P_{CT}$ scores are also provided (targetscan.org). Predictions are also made for homologous 3′ UTRs of other vertebrate species, using either human-centric or mouse-centric 3′-UTR definitions and corresponding MSAs.

## Acknowledgements

We thank the Bioinformatics and Research Computing group at the Whitehead Institute (I Barrasa, B Yuan, Y Huang, and P Thiru) for help implementing improvements to the TargetScan website, A Subtelny for providing insight into positional effects of the miRNA seed, I Ulitsky for initial help with 3P-seq analysis, R Friedman for discussions regarding the computation of $P_{CT}$ parameters, T Tuschl for sharing an unpublished list of the most frequently sequenced human miRNA isoforms, G Agarwal for discussions regarding normalization techniques, G Kudla for help processing the microarray data from the CLASH study, SW Chi and RB Darnell for confirmation of the mRNAs identified as miR-124 targets in their dCLIP study, O Rissland and J Guo for critical reading of the manuscript, and members of the Bartel lab for helpful discussions. This work was supported by a National Science Foundation Graduate Research Fellowship (to VA) and an NIH grant GM067031 (to DPB). DPB is an investigator of the Howard Hughes Medical Institute.

## Additional information

### Funding

| Funder | Grant reference | Author |
| --- | --- | --- |
| National Institutes of Health (NIH) | GM067031 | Vikram Agarwal, George W Bell, Jin-Wu Nam, David P Bartel |
| Howard Hughes Medical Institute (HHMI) | Investigator | David P Bartel |
| National Science Foundation (NSF) | Graduate Research Fellowship | Vikram Agarwal |

The funders had no role in study design, data collection and interpretation, or the decision to submit the work for publication.

### Author contributions

VA, Conception and design, Acquisition of data, Analysis and interpretation of data, Drafting or revising the article; GWB, Overhaul of the TargetScan website, Implementing the improved quantitative model; J-WN, Annotation and quantification of 3′ UTR isoforms, Contributed unpublished essential data or reagents; DPB, Conception and design, Analysis and interpretation of data, Drafting or revising the article

## Additional files

### Supplementary files

• Supplementary file 1. Normalized values for fold changes (log$_2$) of mRNAs detectable in the compendium of 74 sRNA transfection datasets.

• Supplementary file 2. Normalized values for fold changes (log2) of mRNAs detectable in the seven datasets examining the response of transfecting miRNAs into HCT116 cells.

• Supplementary file 3. Genomic coordinates of CLIP clusters that appeared in annotated 3′ UTRs after transfecting miR-124 into HeLa cells.

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
