## [Decision Letter]

Thank you for sending your work entitled “Predicting effective microRNA target sites in mammalian mRNAs” for consideration at *eLife*. Your article has been favorably evaluated by James Manley (Senior editor) and two reviewers, one of whom, Elisa Izarralde, is a member of our Board of Reviewing Editors.

The two experts who evaluated the manuscript were extremely positive about the work and we are very pleased to inform you that your article has been accepted for publication in *eLife* with minor text changes that are indicated below.

We pasted some of the referees' comments, because they were exceptionally positive and you and your co-workers will be pleased to know that the work has been very well received:

“This is very thoughtful and well conducted study to derive an improved quantitative model of canonical microRNA targeting model. The authors show convincingly that non-canonical miRNA-mRNA interactions exist, but seem to be non-functional in repression of their target transcripts.....Using 14 selected features, multiple linear regression models were trained on the data. The derived context++ model outperforms the most recent versions of twelve in silico miRNA target prediction tools.

This manuscript is very timely and of interest to a broad scientific readership.”

“This manuscript represents an extraordinary tour de force and its implementation through TargetScan will be highly significant to the broad field of gene expression regulation and provides a valuable resource to the scientific community.”

The minor text changes suggested by the reviewers are:

The authors should cite the recent structural work on human Ago2 (Schirle et al. Science 2014) and discuss their findings in the context of this work, which supports canonical miRNA interactions.

In regards to context-dependency of miRNA interactions the authors should cite Erhard et al. Genome Research 2014.

In the third and fourth paragraphs of the subsection “Inefficacy of recently reported non-canonical binding sites”. How can the atypical shape of the curve be explained?

---

## [Author Response]

*The authors should cite the recent structural work on human Ago2 (Schirle et al. Science 2014) and discuss their findings in the context of this work, which supports canonical miRNA interactions*.

This work is now cited and discussed in the Introduction (second paragraph) and the Discussion (twelfth paragraph1152–1162).

*In regards to context-dependency of miRNA interactions the authors should cite Erhard et al. Genome Research 2014*.

[53] is now cited in a new paragraph discussing context dependency (Discussion, second paragraph).

*In the third and fourth paragraphs of the subsection “Inefficacy of recently reported non-canonical binding sites”. How* can *the atypical shape of the curve be explained*?

We occasionally observe these and other atypical shapes in cumulative distributions. Because they are not reproducible in independent datasets, we have no reason to think that they have a biological explanation and instead attribute them to chance.